# Young Shoots of Red Beet and the Root at Full Maturity Inhibit Proliferation and Induce Apoptosis in Breast Cancer Cell Lines

**DOI:** 10.3390/ijms24086889

**Published:** 2023-04-07

**Authors:** Ewelina Piasna-Słupecka, Teresa Leszczyńska, Mariola Drozdowska, Kinga Dziadek, Barbara Domagała, Dominik Domagała, Aneta Koronowicz

**Affiliations:** 1Department of Human Nutrition and Dietetics, Faculty of Food Technology, University of Agriculture in Krakow, Balicka 122, 31-149 Krakow, Poland; 2Department of Horticulture, Faculty of Biotechnology and Horticulture, University of Agriculture in Krakow, Al. 29 Listopada 54, 31-425 Krakow, Poland

**Keywords:** young shoots, beetroot, digestion and absorption, breast cancer cells, apoptosis

## Abstract

Modern medicine is struggling with the problem of fully effective treatment of neoplastic diseases despite deploying innovative chemotherapeutic agents. Therefore, undertaking cancer-prevention measures, such as proper eating habits, should be strongly recommended. The present research aimed to compare the effects of juice from young shoots of beetroot compared to juice from root at full maturity on human breast cancer and normal cells. The juice from young shoots, both in the native and digested form, was most often a significantly stronger inhibitor of the proliferation of both analyzed breast cancer cell lines (MCF-7 and MDA-MB-231), compared to the native and digested juice from red beetroot. Regardless of juice type, a significantly greater reduction was most often shown in the proliferation of estrogen-dependent cells (MCF-7 line) than of estrogen-independent cells (MDA-MB-231 line). All analyzed types of beetroot juice and, in particular, the ones from young shoots and the root subjected to digestion and absorption, exerted an antiproliferative and apoptotic effect (pinpointing the internal apoptosis pathway) on the cells of both cancer lines studied. There is a need to continue the research to comprehensively investigate the factors responsible for both these effects.

## 1. Introduction

Cancers pose one of the most critical health problems and have been classified as chronic non-communicable diseases. Despite extensive research conducted in oncotherapy and the dynamic development of innovative diagnostic and therapeutic methods, the incidence of various types of cancer is still increasing. The main reason behind this alarming trend, despite the improving living conditions, is the aging of the population and deteriorating quality of environmental conditions, related mainly to high pollution. Improper lifestyle habits are also crucial in this case, including, among others, ill-balanced diet or exposure to stress [1,2].

Despite the progress in treatment and the use of new innovative chemotherapeutics, modern medicine is still struggling with the problem of fully effective therapy. Therefore, there is a strong need not only to deploy the existing knowledge to develop and synthesize new, potential chemotherapeutics that would be effective against cancer cells but also to place an appropriate emphasis on prevention, such as proper nutritional habits entailing the consumption of large amounts of vegetables and fruits, which have been proven to reduce the risk of cancer development [3].

Such prophylactic measures could be based on new dietary components rich in antioxidants and other biologically active substances. However, searching for compounds with comprehensive anticancer activity and low cytotoxicity against normal cells remains a continuous challenge.

Due to the justified need to support the treatment of diseases, including cancer, with non-pharmacological methods, there is a growing interest in raw materials of plant origin, especially “folk” plants which could be successfully used in prevention. One such plant may be beetroot (*Beta vulgaris* L.), a commonly grown vegetable, especially in Poland.

Most of the research papers addressing beetroot concern mainly its mature vegetative form, but there is a lack of comprehensive studies on the health-promoting properties of this vegetable at an early stage of development, such as young shoots.

Despite the dynamic developments in medicine, knowledge about the molecular mechanisms of the biological activity of beetroot, and even more young shoots, is insufficient. In our previous studies, we observed that young shoots of beetroot (*Beta vulgaris* L.), as the vegetative part of the plant in the intensive growth phase, had significantly higher contents of total protein, fats, and total mineral compounds in the form of ash, compared to the root at full maturity. More importantly, young shoots of red beet were a richer source of total polyphenols having anti-carcinogenic properties and showed higher antioxidant activity [4].

Therefore, due to the differences found in the composition of red beet at different, extreme stages of vegetation, the main goal was to compare the potential anti-carcinogenic effects of juices made of young shoots and beetroot at full maturity on selected lines of human mammary gland cancer cells in vitro.

The present study was preceded by a complementary analysis of individual fractions of one of the most important groups of biologically active compounds, i.e., polyphenols.

To the best of our knowledge, the polyphenolic profile of the juice from young shoots of the beetroot as well as the mechanisms of cell death induced by the juices subjected to in vitro gastrointestinal digestion and absorption have been evaluated for the first time in this study.

## 2. Results

### 2.1. Polyphenolic Compounds

Phenolic acids and flavonoids were identified in the juices obtained from young shoots and root of red beet. Among the phenolic acids from the hydroxybenzoic group, significantly (*p* ≤ 0.05) higher concentrations of gallic acid (5.51-fold), 4-hydroxybenzoic acid (14.11-fold), vanillic acid (12.08-fold), and syringic acid (22.39-fold) were determined in the juice from young shoots compared to juice made from the root. In the group of hydroxycinnamic acids, the juice made from young shoots had significantly (*p* ≤ 0.05) higher concentrations of chlorogenic (1.31-fold), caffeic (10.9-fold), *p*-coumaric (3.05-fold), ferulic (37.27-fold) and sinapinic (4112.4-fold) acids compared to the juice made from root (Table 1, Figure 1 and Figure 2).

Among the flavonoids, significantly (*p* ≤ 0.05) more catechins (2.97-fold) and epicatechins (13.43-fold) from the class of flavanols, rutin from the class of flavonols (1596.64-fold), and hesperidin from the class of flavones (629.72-fold) were assayed in the juice from young shoots of beetroot compared to the juice from root (Table 1, Figure 1 and Figure 2).

Significantly (*p* ≤ 0.05) more carnosol (30.70-fold) and carnosolic acid (5.17-fold) were also determined in the juice from young shoots. Sinapinic acid was found to be among the major compounds of the juice from young shoots of beetroot, whereas catechin was also found in substantial amounts in the juice made from the root.

Kaempferol, myricetin, isorhamnetin, luteolin, apigenin, rosmarinic acid, quercetin, and hispidulin were determined only in young shoots, while naringin was found only in the root.

The comparison of the composition of the tested material shows that statistically significant (*p* ≤ 0.05) differences occurred in concentrations of all identified polyphenolic compounds (Table 1). The total content of polyphenols determined in the juice from young shoots was significantly (*p* ≤ 0.05) higher compared to their content in the root of red beet (Table 1, Figure 1 and Figure 2).

### 2.2. Cell Proliferation

The inhibition of proliferation of MCF-7 and MDA-MB-231 cancer cell lines progressed both with an increasing addition of the native type of juice to the medium (1:9–5:5 *v*/*v*) and, in most cases, with extended incubation time (Table 2, Appendix A).

The addition of the digested juice from young shoots into the MCF-7 cancer cell culture statistically significantly (*p* ≤ 0.05) inhibited cell proliferation by 27.38% after 24-h incubation; further 48-h and 72-h incubation caused successive inhibition compared to the control sample by 33.03% and 58.83%, respectively (Table 2 and Appendix A). In turn, the addition of the digested root juice into the MCF-7 cancer cell culture resulted in a statistically significant (*p* ≤ 0.05) reduction of cell proliferation by 22.97% after 24-h incubation.

Continuation of the experiment for the next two days resulted in a significant (*p* ≤ 0.05) progressive inhibition of the proliferation of the cell line studied, i.e., inhibition by approximately 30.58% and 63.45% after 48 h and 72 h of incubation, respectively (Table 2 and Appendix A).

After 24 h, 48 h, and 72 h of incubation of MDA-MB-231 cancer cells with the digested juice from young shoots of beetroot, their proliferation was significantly (*p* ≤ 0.05) inhibited, by 4.76%, 9.11% and 13.19%, respectively (Table 2 and Appendix A).

The juice made of beetroot root and subjected to gastrointestinal digestion and absorption in vitro significantly (*p* ≤ 0.05) inhibited the proliferation of the MDA-MB-231 cancer cell line after 72 h of incubation by 9.81% (Table 2 and Appendix A).

### 2.3. Cytotoxicity

The juice made of young shoots and root of beetroot subjected to gastrointestinal digestion and absorption in vitro had no cytotoxic effect on MCF-7 and MDA-MB-231 cell lines after their 24-h incubation (*p* > 0.05) (Table 3 and Appendix A). Statistically significant (*p* ≤ 0.05) increased toxicity of the studied juices, but not exceeding 10%, was determined on the second day of the experiment in the case of both analyzed cell lines (Table 3 and Appendix A). After 72 h of incubation with MCF-7 and MDA-MB-231 cell lines, a further increase was observed in the cytotoxicity of both the digested juices (Table 3 and Appendix A).

### 2.4. Selected Cellular Parameters Analyzed Using a Flow Cytometer

#### 2.4.1. Apoptotic Activity

Flow cytometry with the Muse^®^ Annexin V and Dead Cell Assay was performed to investigate the mode of the studied juices-induced cell death in MCF-7 and MDA-MB-231 breast cancer cell lines (representative plots in Appendix A). The analysis of the obtained results confirmed the ability of the tested material to induce apoptosis in the breast cancer cell lines MCF-7 and MDA-MB-231. After 48-h incubation, a statistically significant (*p* ≤ 0.05) increase was observed in the fraction of apoptotic cells (Table 4). The strongest significant (*p* ≤ 0.05) effect was found for the digested and absorbed juices from young shoots and root of beetroot. The fraction of apoptotic cells was over 55% in both cases compared to the control sample of MCF-7 as well as 82.20% and 69.75%, respectively, in the case of the MDA-MB-231 line (Table 4). The native juice from young shoots and the root also caused cell apoptosis, which was significantly (*p* ≤ 0.05) more intense than in the control sample.

#### 2.4.2. BCL-2 Activity

The impact of the tested material on the levels of antiapoptotic BCL-2 protein in human breast cancer cell lines was assessed using the Muse™ Bcl-2 Activation Dual Detection Assay (Merck Millipore, Billerica, MA, USA) (representative plots in Appendix A). Both types of juices were able to inactivate the BCL-2 protein in MCF-7 and MDA-MB-231 cancer cell lines. After 48-h incubation, a statistically significant (*p* ≤ 0.05) increase was determined in the fraction of cells in which the antiapoptotic protein was inactivated compared to the control sample. The strongest significant (*p* ≤ 0.05) effect was noted for the juices made from young shoots of beetroot, both for the juice subjected to the process of digestion and absorption as well as the native one. The fraction of these cells was over 90% in both cases in MCF-7 and 97.50 and 83.50%, respectively, in the MDA-MB-231 cell line (Table 5).

#### 2.4.3. Multicaspase Activity

Following the assessment of Bcl-2 levels, we proceeded with evaluating the profiles of the activity of other proteins involved in cell death, namely caspases 1, 3, 4, 5, 6, 7, 8 and 9, after treatment with both types of juices, using the Muse^TM^ MultiCaspase Assay (Merck Millipore, Billerica, MA, USA) (representative plots in Appendix A). The obtained results confirmed the ability of the tested material to activate caspases in breast cancer cell lines MCF-7 and MDA-MB-231. The strongest significant (*p* ≤ 0.05) effect on the MCF-7 cell line was recorded for the juice from young shoots of beetroot, both the one subjected to the digestion and absorption process as well as the native juice. The affected cells accounted for 72.70% and 67.05% of the total, respectively (Table 6). In turn, in the MDA-MB-231 cell line, the strongest significant (*p* ≤ 0.05) effect was noted for the digested juice made of young shoots and the digested juice made of root. The affected cells constituted 58.25% and 73.60%, respectively (Table 6).

### 2.5. mRNA Expression Analysis

The study showed that the analyzed juices significantly increased or decreased the expression of most of the genes tested. Real-time PCR analysis showed that in the case of the MCF-7 cell line, the native juice from young beetroot shoots significantly (*p* ≤ 0.05) increased the expression of the following genes: *AIFM1* (2.05-fold), *AKT1* (1.26-fold), *APAF1* (4.12-fold), *BAD* (2.03-fold), *BID* (1.81-fold), *CASP7* (5.07-fold), *CASP8* (4.61-fold), *MYC* (1.58-fold), *NFKB1* (3.17-fold), and *TP53* (1.70-fold) compared to the control sample (Table 7 and Appendix A). In the same cell line, the juice made of root significantly (*p* ≤ 0.05) enhanced the expression of the following genes: *AKT1* (1.19-fold), *CASP7* (1.96-fold), *MYC* (1.48-fold) and *NFKB* (1.40-fold) relative to the control sample (Table 7 and Appendix A).

Real-time PCR analysis showed that in the MCF-7 cell line incubated in the medium with the addition of the juice from young shoots of beetroot, subjected to the process of digestion and absorption in a digestive tract model in vitro, there was a significant (*p* ≤ 0.05) increase in the expression of the following genes compared to the control group: *AIFM1* (2.17-fold), *APAF1* (5.05-fold), *BAD* (1.84-fold), *CASP8* (3.16-fold), *DIABLO* (2.55-fold), *MYC* (1.89-fold), and *TP53* (1.55-fold). At the same time, a significant (*p* ≤ 0.05) suppression was observed in the expression of *AKT1* (1.97-fold) and *NFKB1* (1.40-fold) genes (Table 6 and Appendix A). In contrast, the juice made of root of beetroot, subjected to the digestion and absorption process, significantly (*p* ≤ 0.05) enhanced the expression of *MYC* (1.78-fold) and *TP53* (1.31-fold) genes compared to the control group. At the same time, the discussed cells showed a significant (*p* ≤ 0.05) decrease in the expression of the *AKT1* (2.51-fold) and *BCL-2* (1.68-fold) genes (Table 7 and Appendix A).

Real-time PCR analysis showed no expression of the *CASP3* gene in cancer cells of the MCF-7 line in any of the analyzed experimental samples (Table 7).

In the case of the MDA-MB-231 cell line cultured in a medium with native juice of young beetroot shoots, the real-time PCR demonstrated a significant (*p* ≤ 0.05) increase in the expression of such genes as *AIFM1* (2.14-fold), *APAF1* (2.45-fold), *BAD* (1.24-fold), *DIABLO* (1.99-fold), *NFKB1* (2.02-fold), and *TP53* (2.81-fold) compared to the control sample. At the same time, a significant (*p* ≤ 0.05) suppression was observed in *CASP8* (5.01-fold) and *FADD* (1.92-fold) gene expression in these cells (Table 7 and Appendix A).

Likewise, the transcript expression levels of *AIFM1* (7.75-fold), *APAF1* (2.89-fold), *BAD* (1.34-fold), *BBC3* (2.19-fold), *BID* (2.91-fold), *DIABLO* (2.25-fold), *MYC* (3.54-fold), *NFKB* (6.40-fold), and *TP53* (2.60-fold) were increased upon the treatment with the native root juice compared to the control. At the same time, native root juice caused a significant (*p* ≤ 0.05) suppression in *AKT1* (1.27-fold) and *FADD* (2.65-fold) gene expression (Table 7 and Appendix A).

In contrast, the juice made of young shoots of beetroot, subjected to the process of digestion and absorption, significantly (*p* ≤ 0.05) enhanced the expression of genes: *AIFM1* (3.72-fold), *APAF1* (2.10-fold), *BAD* (4.21-fold), *BBC3* (2.65-fold), *BID* (1.57-fold), *CASP8* (3.38-fold), *DIABLO* (1.43-fold), *MYC* (1.45-fold), *NFKB1* (5.48-fold), and *TP53* (1.45-fold) compared to the control group, while significantly (*p* ≤ 0.05) suppressing the expression of *AKT1* (2.23-fold) and *FADD* (5.30-fold) genes in the MDA-MB-231 cell line (Table 7 and Appendix A).

On the other hand, the same cells incubated in the medium with the addition of digested beetroot juice from the root showed a significant (*p* ≤ 0.05) increase in the expression of the following genes: *AIFM1* (3.85-fold), *APAF1* (2.66-fold), *BAD* (3.29-fold), *CASP8* (2.44-fold), *DIABLO* (2.50-fold), *MYC* (1.38-fold), *NFKB1* (3.96-fold), and *TP53* (1.35-fold) compared to the control group. At the same time, a significant (*p* ≤ 0.05) reduction in the expression of the *AKT1* (2.16-fold) and *FADD* (3.00-fold) genes was observed in these cells (Table 7 and Appendix A).

### 2.6. Protein Expression Analysis

Western blot analysis showed a significant (*p* ≤ 0.05) increase in the expression of proteins such as cytochrome c (2.10-fold) and caspase 8 (3.03-fold) in the MCF-7 cell line cultured in a medium with the addition of native juice from young beetroot shoots. At the same time, the RIP protein expression decreased significantly (*p* ≤ 0.05) in the cells of the discussed cell line (3.04-fold) (Figure 3 and Appendix A).

In the MCF-7 cell line, incubated in a medium with native juice from root of beetroot, a significant (*p* ≤ 0.05) increase was observed in the expression of such proteins as cytochrome c (2.03-fold), Smack/DIABLO (1.4-fold), HtrA2/Omi (2.43-fold), caspase 7 (5.21-fold), caspase 8 (2.77-fold), and p38 (2.39-fold) compared to the control group of cells. At the same time, a significant (*p* ≤ 0.05) decrease in RIP protein expression was noted (1.45-fold) (Figure 3 and Appendix A).

The Western blot analysis showed a significant (*p* ≤ 0.05) enhancement in the expression of such proteins as cytochrome c (2.95-fold), Smack/DIABLO (2.32-fold), HtrA2/Omi (2.45-fold), caspase 7 (5.84-fold), caspase 8 (5.65-fold), p53 (3.3-fold), and cleaved PARP (2.7-fold) compared to the control group under the influence of juice made of young shoots of beetroot, subjected to the process of digestion and absorption in vitro in the MCF-7 cell line. This juice also caused a significant (*p* ≤ 0.05) decrease in RIP protein expression (2.05-fold) (Figure 3 and Appendix A).

Our results also showed a significant (*p* ≤ 0.05) increase in the expression of the following proteins: cytochrome c (1.78-fold), Smack/DIABLO (2.26-fold), HtrA2/Omi (1.99-fold), caspase 7 (2.47-fold), caspase 8 (4.37-fold), p53 (2.34-fold), cleaved PARP (1.85-fold), and p38 (2.35-fold) compared to the control group, as a result of MCF-7 cell line treatment with juice made of the root of beetroot, subjected to the process of digestion and absorption in vitro. At the same time, the use of this juice resulted in a significant (*p* ≤ 0.05) reduction in RIP protein expression (3.16-fold) (Figure 3 and Appendix A).

In turn, in the MDA-MB-231 cell line, Western blot analysis showed that the effect of native juice from young beet shoots added to the culture medium, resulted in a significant (*p* ≤ 0.05) increase in the expression of Smack/DIABLO (1.67-fold) and p53 (1.78-fold) proteins compared to the control group. At the same time, the use of this juice resulted in a significant (*p* ≤ 0.05) reduction in the expression of cytochrome c (1.83-fold), caspases 7 (1.78-fold), caspases 8 (1.79-fold), and RIP (8.56-fold) when compared to the control cells (Figure 4 and Appendix A).

The effect of the native juice made of the root of beetroot on the MDA-231-MB cell line was a significant (*p* ≤ 0.05) increase in the expression of the following proteins: cytochrome c (1.49-fold), Smack/DIABLO (1.88-fold), p53 (7.41-fold), and p38 (2.58-fold) compared to the control cells. At the same time, the effect of this juice was manifested in a significant (*p* ≤ 0.05) suppression in the expression of HtrA2/Omi (7.44-fold), caspase 7 (2.12-fold), caspase 8 (24.19-fold), and RIP (1.68-fold) proteins (Figure 4 and Appendix A).

In the case of the MDA-231-MB cell line, addition of the juice made of young shoots of beetroot, subjected to the process of digestion and absorption, significantly (*p* ≤ 0.05) enhanced the expression of such proteins as cytochrome c (2.49-fold), Smack/DIABLO (1.5-fold), p53 (3.3-fold), caspase 7 (1.35-fold), caspase 8 (1.89-fold), cleaved PARP (2.34-fold), and p38 (2.27-fold), compared to the control cells, while significantly (*p* ≤ 0.05) suppressing the expression of HtrA2/Omi (1.64-fold) and RIP (1.11-fold) proteins (Figure 4 and Appendix A).

The juice made of root of beetroot, subjected to the process of digestion and absorption, on MDA-231-MB cell line caused a significant (*p* ≤ 0.05) increase in the expression of the following proteins: cytochrome c (2.26-fold), Smack/DIABLO (2.39-fold), p53 (38.21-fold), caspase 8 (1.31-fold), and p38 (2.91-fold) compared to the control group. At the same time, it triggered a significant (*p* ≤ 0.05) decrease in the expression of HtrA2/Omi (1.75-fold) and RIP (1.35-fold) proteins (Figure 4 and Appendix A).

## 3. Discussion

At the beginning of the 20th century, research in food science and nutrition concerned, among others, identifying diseases directly or indirectly related to deficiency in or an excess of energy, or to a specific nutrient in the human diet. Later, the demand for individual nutrients was determined. Currently, much emphasis is placed on defining the role of diet composition in maintaining good health, including intellectual and physical ability, and in reducing the risk of developing chronic non-communicable diseases, including cancer [5,6].

The widely recognized threat of the growing incidence of diet-related diseases and of the risk of developing them has resulted in the search for functional foods or novel dietary elements that could be deployed not only in prevention but also in supporting therapy, notably in respect of cancers. The available treatment methods are still ineffective, and research results confirm the rapidly growing resistance of cancer cells to the current chemotherapy methods. Therefore, there is a need to also search for new effective substances of natural origin [7]. The results of population and experimental studies generally confirm the health-promoting effect of following diets rich in products of plant origin [8,9,10]. Although the exact mechanisms are still not fully understood, a potential consumer interested in health promotion and protection will look for food with a preventive effect proven in research.

According to recommendations, vegetables and fruits are crucial elements, and should account for half of the total amount of the human diet. They provide valuable nutrients and non-nutrients with health-promoting properties [11]. According to the recommendations of the World Health Organization and the National Center for Nutritional Education in Poland, the daily intake of fruit and vegetables should be at least 400 g. The beneficial effects of raw materials of plant origin include their antioxidant, anti-inflammatory, antimutagenic, and anti-carcinogenic properties, inhibiting both the initiation of cancer development and its progression.

Beetroot (*Beta vulgaris* L.) is one of the vegetables with significant biological potential and health-promoting properties. It contains many components, including polyphenols, vitamins (folates, vitamin C), minerals (potassium, iron, calcium), and betalains (betanin) with multiple proven activities, including anti-carcinogenic activity [12]. This vegetable is widespread in Poland and Europe and can be consumed in various forms. However, due to the long development period and reaching the phase of full consumption maturity, it seems reasonable to introduce plant components to the diet whose growth time would be maximally shortened and which could be suitable carriers of bioactive substances [13]. Such plants are called young shoots/leaves in Polish academia and microgreens in English-written studies [14].

In recent years, young shoots have been gaining in popularity as food components due to the usually higher content of nutrients and non-nutrient ingredients with proven health-promoting properties compared to their counterparts—mature plants with different sensory characteristics. In the intensive development phase, young shoots are characterized by greater contents of key components and higher enzymatic activity compared to a mature plant [4,15]. Young shoots and plant sprouts have many common features, but due to their small size resulting from a shorter growth cycle, obtaining sufficiently large biomass requires appropriate inputs and agro-technical conditions [16].

It is worth emphasizing that the study of the product, as opposed to the individual constituents, makes it possible to demonstrate their combined interfering (synergistic/antagonistic) effect in the subject area. This gives an advantage over the results of studies evaluating the impact of single, isolated compounds, although these are necessary to understand the mechanisms of their action. At the same time, as has been proven, the effect of individual compounds can be utterly different from the impact of their combined action [17].

In addition, the use of in vitro digestion and absorption processes in research studies allows the bioavailability of the compounds contained in the product to be taken into account and therefore allows for a more realistic assessment of their combined effect on the body cells.

No comparative results were found in the available literature for the effectiveness of the above-mentioned products before and after digestion on the proliferation and changes in selected cellular parameters responsible for the apoptosis of cancer cells. In addition, no attempt has ever been made to compare the effectiveness of the analyzed beetroot material in relation to human breast cancer cells of the hormone-dependent (MCF-7) and hormone-independent (MDA-MB-231) lines.

The results of determinations of the proximate composition, total polyphenol content, and antioxidant activity of young shoots and fully-mature roots of red beet (*Beta vulgaris* L.) are presented in our previous publication [4]. The young shoots of beetroot, as the vegetative part of the plant in the stage of intensive growth, are characterized by a significantly higher content of total protein, fats and total mineral compounds in the form of ash compared to the root in the full maturity phase. In addition, young shoots of beetroot are a richer source of total polyphenols and show higher antioxidant activity.

Other authors [18] also generally found a higher content of nutrients and some non-nutrients in young shoots compared to the mature plant. However, Xiao et al. [15] did not obtain such unambiguous results, showing that it was the ripe vegetable that had higher contents of certain compounds.

### 3.1. Polyphenolic Profile

Polyphenols constitute a structural class of organic chemical compounds possessing at least two hydroxyl groups attached to an aromatic ring. The number and characteristics of these structures determine their physical, chemical, and biological properties. Polyphenols in plants are responsible for color, sensory properties and above all, they exert a chemopreventive effect by transferring protons from hydroxyl groups to reactive oxygen species. Due to their antioxidant, anti-inflammatory, anti-mutagenic and anti-carcinogenic properties, they can prevent chronic non-communicable diseases. Polyphenols can be classified according to their chemical structure and divided into several categories, including phenolic acids, flavonoids, lignans, and stilbenes.

The HPLC analysis performed in the present study enabled detection of phenolic acids and flavonoids in the juice from young shoots of beetroot. Their contents were compared with those obtained in the juice from the root in the full maturity phase. A significantly higher total content of polyphenols was determined in the juice from young shoots compared to the juice from the root of the analyzed cultivar Rywal. In young shoots of beetroot, the major compound was sinapinic acid, from the hydroxycinnamic group; in the root, however, it was catechin, which belongs to the class of flavanols (Table 1). No anthocyanins were detected in the studied material, due to the lack of the anthocyanin synthetase enzyme involved in the final stage of the pigment formation [19].

In this study, out of 22 polyphenolic compounds identified in the juice from young shoots of beetroot, hydroxybenzoic acid, hydroxycinnamic acid, flavanols, flavonols, and flavones accounted for 3.73%, 24.67%, 6.27%, 22.36%, and 21.30% of the total polyphenols.

The available literature lacks data on the content of determined polyphenol fractions in young shoots of beetroot. For this reason, the works cited below concern other species of young shoots, including those from the *Chenopodiaceae* family, which encompasses *Beta vulgaris*. In many plants belonging to the mentioned family, Kyriacou et al. [20] quantified 28 types of phenolic compounds, such as hydroxycinnamic acids, flavonol glycosides, as well as flavones and flavonol glycosides, representing 7.6%, 24.8%, and 67.6% of total polyphenols.

In studies on young shoots of kohlrabi, pak choi, and coriander, changes in the content of polyphenols were observed, depending on the type of substrate (natural vs. synthetic fiber). Chlorogenic acid and quercetin were found to be the major compounds among the 20 identified phenolics.

Hydroxycinnamic acid and its derivatives, flavonol glycosides and flavone glycosides, accounted for 49.8%, 48.4% and 1.8% of total polyphenols. The results obtained in our study are lower than those cited above.

According to Kyriacou et al. [21], the content of polyphenols in young shoots and growing conditions is also influenced by species variability.

On the other hand, in beetroot root, out of the 16 identified polyphenolic compounds, hydroxybenzoic acid, hydroxycinnamic acid, flavanols, flavonols, and flavones accounted for 24.83%, 8.48%, 51.31%, 0.48%, and 1.33% of the total content of determined polyphenols.

Using the HPLC technique, Kale et al. [22] demonstrated the presence of several polyphenolic compounds in the beetroot root, such as gallic acid, *p*-coumaric acid, ferulic acid, cinnamic acid, and catechol, but in higher amounts compared to the results presented in this paper.

Values similar to those obtained by Kale et al. [22] were also presented by Vulić et al. [23], who, in their experiments, additionally determined flavonoids, such as betagarin, betavulgarine, cochlyophyllin A, and dihydroisoramnetin.

In turn, Jastrebova et al. [24] determined the following polyphenols in the *Beta vulgaris* root: catechin, quercetin, *p*-coumaric acid, sinapinic acid, syringic acid, caffeic acid, and chlorogenic acid. The results reported in the cited work were also higher than those determined in the present study. The studies by Amani et al. [25], analyzing the profile of polyphenols in the beetroot root, showed a higher content of hydroxybenzoic acid, rutin, hesperidin, kaempferol, naringin, apigenin, and isorhamnetin, compared to those presented here. In addition, these researchers detected ellagic acid. Lim [26] also determined other flavonoid compounds in beetroot: 3,5-dihydroxy-6,7-methylenedioxyflavanone, 5-hydroxy-6,7methylenedioxyflavone, 2,5-dihydroxy-6 and 7-methylenedioxy isoflavone. In turn, Georgiev et al. [27] showed the presence of 4-hydroxybenzoic acid, chlorogenic acid, caffeic acid, ferulic acid, vanillic acid, catechin, and epicatechin in beetroot of the Detroit Dark Red variety as well as rutin, again in amounts higher than those found in the present study.

Studies with leaves (chard) of *Beta vulgaris* L. var. cicla showed significant amounts of hydroxybenzoic and hydroxycinnamic acid derivatives. Epicatechin, catechin, rutin, vanillic acid, *p*-coumaric acid, protocatechuic acid, caffeic acid, syringic acid and proline were determined among the phenolic acids [28].

The content of polyphenols may vary depending on the variety of beetroot [29] and on the analyzed site in the root cross-section [30].

Previous studies have confirmed the antiproliferative effect of polyphenols on T47D, MCF-7, and MDA-MB-231 breast cancer cell lines in a time- and dose-dependent manner [31]. These compounds affect cellular mechanisms and molecules related to carcinogenesis. Selected polyphenolic compounds have been found capable of suppressing the activity of transcription factors, such as NF-κB and AP-1 [32]. These protein complexes control many genes that regulate proliferation and apoptosis, and disturbances in their pathways can lead to carcinogenesis. It seems that the inhibitory effect of polyphenols on the aforementioned transcription factors may be due to their antioxidant properties because reactive oxygen species can activate both NF-Κb and AP-1. The anti-proliferative and pro-apoptotic effect of polyphenols is also explained by the inhibition of critical proteins involved in signal transduction pathways from the cell membrane to the cytoplasm and nucleus and in regulation of the cell cycle, as well as in apoptosis through the initiation of caspase-3, activation of p53 protein and of factors affecting cell proliferation and differentiation [33]. It must be highlighted that polyphenols exert a stronger effect on cancer cells than on normal cells [34].

### 3.2. Cell Proliferation

The analysis of the obtained results showed that the juices from the young shoots and root of red beet, both in their native form and subjected to digestion and absorption, inhibited the proliferation of the MCF-7 and MDA-MB -231 breast cancer cell lines (Table 2) while not affecting the proliferation of MCF-12A normal cell line. Due to the need for effective agents that inhibit the proliferation of cancer cells and at the same time do not show a similar effect on normal cells of the body, these data seem promising.

In addition, it was observed that the juice from young shoots, both in the native form and after digestion and absorption, in most cases reduced the proliferation of cells of both cancer lines to a significantly greater extent than the juice made from root in the full maturity phase.

The difference in the efficacy of the analyzed material on both breast cancer cell lines should also be emphasized; the effect was usually significantly higher in relation to estrogen-dependent cells of the MCF-7 line.

The available literature provides no information on the effect of young shoots of beetroot on the proliferation of cancer cells.

In this study, analyses were carried out using the juice obtained from the native plant material and also subjected to a digestion and absorption process in a model digestive tract in vitro. On the other hand, the papers cited below report on the effects of the direct extract or single isolated components.

Reddy et al. [35] compared the effects of natural pigments, including betanin, as well as their mutual interactions, on the possibility of inhibiting the growth of cancer cells. In a 48-h experiment, the authors observed growth inhibition of breast cancer cells of the MCF-7 line treated with betanin. Research by Kapadia et al. [36] showed that beetroot extract strongly inhibited the proliferation of cancer cells, including the breast cells analyzed in this study (line MCF-7). Another study demonstrated a synergistic anti-proliferative effect of doxorubicin and beetroot extract on breast and prostate cancer cell lines [37]. Nowacki et al. [38] observed suppressed proliferation of MCF-7 and MD-MB-231 breast cancer cell lines under the influence of a mixture of betanin/isobetanin, with no significant effect on the normal HUV-EC-C cell line. In turn, Das et al. [39] demonstrated the chemotherapeutic effect of beetroot extract in combination with doxorubicin on the viability of MDA-MB-231 cancer cell line.

It was shown that betanin, isolated from red beets, depending on the dose, inhibited the proliferation of the lung cancer cells NCI-H460, the CNS (central nervous system) line, as well as gastric AGS and large intestine HCT-116 lines (gastrointestinal tract) [40]. Another study showed an antiproliferative effect of beetroot extract against breast cancer cells (MCF-7) and additionally a beneficial effect on lung (A-549), kidney (Caki-2), prostate (PC-3), pancreas (Panc-1), and chronic myelogenous leukemia (K-562) cancer cell lines [41]. In addition, betanin and betaine, extracted from beetroot, exhibited antiproliferative activity against liver cancer cells of the HepG2 line [42]. Inhibition of proliferation under the influence of betalains extracted from beetroot root was also observed in studies on bladder cancer cells of the T24 line [43]. In the latest 48-h experiment, Saber et al. [44] showed the inhibitory effect of betanin and beetroot extract on HT-29 colorectal cancer cells and Caco-2, as measured by proliferative capacity.

As described above, several in vitro studies have confirmed the effectiveness of an extract or individual phytonutrients most commonly isolated from beetroot root in reducing the proliferation of various types of cancer cells.

In addition, the literature provides the results of research on the antiproliferative effects of young barley [45] or young shoots of cruciferous vegetables [46]. Interesting results were obtained by Drozdowska et al. [47], who assessed the effect of the juice from young shoots of white head cabbage (*Brassica oleracea* var. *capitata f. alba*) on the proliferation of the MCF-7 breast cancer cell line. This study showed a decrease in cell viability depending on the analyzed juice’s concentration and duration of action. In subsequent research, Drozdowska et al. [48] compared the effectiveness of young shoots of red head cabbage (*Brassica oleracea* var. *capitata f. rubra*) with the effect of the plant in full maturity. The authors demonstrated a tendency for the inhibition of the proliferation of DU-145 prostate cancer cell line, both under the influence of juice from young shoots of red head cabbage and from the mature form of this vegetable.

The results obtained in this study concerning the impact of beetroot juice in the full maturity phase, undigested and digested, are generally consistent with the results reported by other authors quoted above who, as already emphasized in this paper, investigated the effectiveness of direct extracts or single components isolated from the red beet.

### 3.3. Cytotoxicity

The analysis of lactate dehydrogenase (LDH) activity in the culture medium is one of the methods for assessing the cytotoxicity of the material towards cells in in vitro studies. As a result of cell membrane disintegration, the entire cell contents, together with the enzymes, are released into the culture medium during necrotic death.

The LDH test conducted in the present study demonstrated that the juice from young shoots of beetroot, both in the native form and after digestion and absorption, exhibited higher cytotoxicity against breast cancer cells of MCF-7 and MDA-MB-231 lines than the native and in vitro digested root juice (Table 3).

In addition, attention should be given to the usually significantly lower efficiency of the analyzed material against the cells of the MDA-MB-231 line, caused probably by a more invasive phenotype characterizing this cell line (Table 3).

As in the case of the previous analysis, the available literature lacks the results of studies on the cytotoxic effect of juice from young shoots and beetroot, both before and after digestion, against normal and cancer cells. According to the findings of other authors quoted below, the cytotoxic effect has been tested only in reference to the direct extract or single, isolated compounds, mainly betanin.

Kapadia et al. [36] compared the cytotoxicity of beetroot extract with the effect of an anticancer drug (doxorubicin) against human prostate and breast cancer cells of the PC-3 and of the MCF-7 line, respectively. According to the results, both beetroot extract and doxorubicin showed dose-dependent cytotoxic effects on both cell lines analyzed. Although the cytotoxicity of the beetroot extract was lower compared to doxorubicin, the former inhibited the proliferation rate of PC-3 prostate cancer cells by 3.7% in 3 days and MCF-7 breast cancer cells by 12.5% in 7 days. In later studies, these authors confirmed the cytotoxic effect of beetroot extract on cancer breast cells of the MCF-7 line after 72 h of exposure [37]. In turn, Nowacki et al. [38] observed an increase in the cytotoxicity of betanin against breast cancer cells of the MCF-7 line but at a much lower concentration, probably due to the use of a chemically pure pigment. Das et al. [39] showed significantly reduced viability of MDA-MB-231 cancer cell lines exposed to a mixture of different concentrations of beetroot extract and doxorubicin compared to the effect of the drug itself. 

In conclusion, no toxic effect of the analyzed juices was found on normal cells of the MCF-12A line. However, cytotoxicity was demonstrated in the case of the two highest volume fractions of the juice in the medium for breast cancer cells of the MCF-7 line and using the highest fraction with the MDA-MB-231 cancer cell line.

Due to up to 10% inhibition of proliferation, compared to the control group of cells, found in these studies corresponding to the first toxic dose (EC10) in subsequent studies, a medium with a volume fraction of native juice at 3:7 (*v*/*v*) was used and the incubation time was set at 48 h.

### 3.4. Selected Cellular Parameters Analyzed Using a Flow Cytometer

From the point of view of chemoprevention, the activation of cell death is a prerequisite for arresting the cancer process at an early stage. Many studies that use cancer cell lines as an experimental model focus on the process of apoptosis. Numerous nutritional studies also aim at evaluating products of natural origin in terms of regulating the course of this process. Apoptosis is a desirable programmed process because it allows living organisms to control the number of cells. Although apoptosis and necrosis are different processes from molecular, biochemical, and morphological perspectives, they lead to the elimination of damaged, abnormal, infected, or simply redundant cells [49].

A characteristic feature of early apoptosis is the loss of cell membrane asymmetry and the translocation of phosphatidylserine from the inner to the outer region. As a result, a protein called Annexin V binds to phosphatidylserine instead of the phospholipids. [50].

The flow cytometry analysis performed in the present study enabled detection of early apoptotic changes indicative of apoptosis induction under the influence of native juices, both from young shoots and the root, in both of the analyzed cell lines (Table 4). The action of young shoots (both types) was significantly more intense in the case of the MDA-MB-231 breast cancer cell line. In addition, a significant increase in the fraction of apoptotic cells was demonstrated as a result of the effect of the juice subjected to the simulated digestion and absorption compared to the impact of its native form on both assessed research models (Table 4), which is consistent with the inhibition of the proliferation of both breast cancer cell lines MCF-7 and MDA-MB-231 demonstrated in this study (Table 2). Moreover, compared to digested root juice, significantly higher pro-apoptotic properties of young shoots were observed against the more invasive, estrogen-independent MDA-MB-231 line. Estrogen-dependent breast cancer MCF-7 cells were also characterized by a large increase in the apoptotic fraction, but probably due to the mutation in the coding gene caspase-3, the level of apoptosis induction in these cells was lower [51]. Although studies have indicated that apoptosis may possibly proceed without activation of the most important effector, caspase-3, cysteine proteases play a crucial role in the process of programmed cell death [52]. While some caspases function to initiate the intracellular cascade, others, called effectors, are responsible for cell breakdown by cleaving their structural proteins.

The results presented in this paper, concerning the activation of several caspases simultaneously (caspases-1, -3, -4, -5, -6, -7, -8 and -9) in the examined cancer cells, confirm the induction of the apoptosis process (Table 6). In the cell culture of both MCF-7 and MDA-MB-231 lines, the highest increase in the activity of caspases was observed with juice from young shoots of beetroot subjected to the process of digestion and absorption. This increase was significantly higher for the cells of both research models analyzed when treated with digested root juice (Table 6).

Bcl-2 family proteins are the main regulators of the intrinsic apoptosis pathway. Mitochondrial dysfunction is associated with a change in permeability and loss of mitochondrial potential. This process leads to the formation of pores in the mitochondrial membrane and to the release of cytochrome c and apoptosis-promoting proteins into the cytoplasm [49]. The phosphorylation process regulates the ability of the BCL-2 protein to inhibit apoptosis. Therefore, the phosphorylated form of this protein has an anti-apoptotic function, whereas dephosphorylation triggers pro-apoptotic activity.

Our research demonstrated the inactivation of the anti-apoptotic protein BCL-2 under the influence of the analyzed juices. The effect of native and digested juice from young shoots was significantly more intense compared to the native and digested root juice in both of the analyzed cancer cell lines (Table 5).

The influence of beetroot root extract on the ability to induce apoptosis in breast cancer cells of the MCF-7 line was also proved by Nowacki et al. [38]. Programmed cell death under the influence of the extract from beetroot root, but using a cancer cell line MDA-MB-231, was demonstrated by Das et al. [39]. In addition, the authors mentioned above argued that apoptosis was significantly more intensive in cells of an estrogen-independent cell line as a result of combined action with doxorubicin compared to the effect of the drug alone. In studies on other cancer cells (colon HT-29 and Caco-2 lines), Saber et al. [44] showed that betanin induced apoptosis at a lower concentration than beetroot extract without affecting normal KDR/293 cells.

### 3.5. Expression of Selected Genes and Proteins

In the present study, the most beneficial effect was demonstrated for the juice from young shoots of beetroot, subjected to digestion and absorption in a gastrointestinal tract model in vitro, against breast cancer cells of the MCF-7 and MDA-MB-231 lines, in comparison with undigested and digested root juice.

Digested root juice was also effective but to a significantly lesser extent. Therefore, to explain the mechanism responsible for reducing proliferation, the level of expression of both genes and proteins related to cell apoptosis under the influence of the mentioned most effective factor, as well as digested root juice, was discussed and confirmed in the previously described analyses using flow cytometry.

The available literature lacks information on the profile of changes in the expression of the genes and proteins discussed below in breast cancer cells of both analyzed lines under the influence of the juice, not only from young shoots but also from vegetables in the full maturity phase. On the other hand, the quoted literature items refer to changes in mRNA and/or protein levels under the influence of beetroot root extract, a single isolated compound, or a mixture thereof, often also against other cancer cell lines.

One of the main pro-apoptotic factors is the *TP53* gene, the product of which is the so-called guardian of the genome. As a transcription factor, the p53 protein is involved in the regulation of many cellular processes, in particular in the activation of DNA repair mechanisms, and in their absence, in the introduction of the cell into the path of apoptosis [53].

The present study results show a significantly higher expression of the *TP53* gene, both under the influence of the digested juice from young shoots as well as from the root, in MCF-7 (Appendix A) and MDA-MB-231 (Appendix A) cell lines compared to control cells. The gene expression results were correlated with the protein expression results determined by Western blot (Figure 3 and Figure 4). In addition, the study demonstrated a tendency (Appendix A) for the enhanced expression of the *BBC3* gene, also known as *Puma*, in the MCF-7 cell line and significantly higher expression in the MDA-MB-231 cell line under the influence of digested young shoot juice (Appendix A). In turn, digested root juice did not affect the expression of the above-mentioned gene in any of the analyzed cell models (Table 7).

Nowacki et al. [38] also showed that betanin, isolated from beetroot, induced apoptosis in MCF-7 breast cancer cell line by activating the p53 protein.

The p53 protein plays a key role in the induction of apoptosis, including by regulating the Bcl-2 family proteins when DNA repair is impossible [40]. The Bcl-2 family includes pro-apoptotic proteins (BID, BAX, BAD, BIM, PUMA) and anti-apoptotic proteins (BCL-2 and BCL-XL). Their effect on apoptosis induction depends on the activity and ratio of pro- and anti-apoptotic proteins [54].

In the present research, a significantly increased expression of *BAD* gene was observed in both breast cancer cell lines (Table 7 and Appendix A) and a significantly increased expression of *BID* gene in the MDA-MB-231 cell line (Table 7 and Appendix A) under the influence of digested juice from young beetroot shoots.

In turn, a tendency for enhanced expression of *BAD* in the MCF-7 cell line (Appendix A) and significantly higher expression of this gene in the MDA-MB-231 cell line (Appendix A) was demonstrated as a result of the action of digested root juice. In addition, this juice did not affect the changes in *BID* expression in either of the analyzed research models (Appendix A). The protein products of these genes are usually located in the cytoplasm, but during the induction of apoptosis, they are activated and transferred to the outer mitochondrial membrane.

Nowacki et al. [38] also showed that betanin, isolated from beetroot root, increased the level of BAD protein expression in MCF-7 cell line. No studies have been found in the available literature regarding the impact of products from beetroot on the expression of the *BID* gene and protein.

In this study, we observed a tendency for downregulation of the expression of the *BCL-2* gene, encoding an anti-apoptotic protein, under the influence of digested juice of young shoots and a significant decrease in the level of this gene as a result of the action of digested root juice (Table 7 and Appendix A) in the MCF-7 cell line. The RT-qPCR conducted in the present study partially confirmed earlier observations made using a flow cytometer.

Opposite results were published by Das et al. [39], who reported a decrease in the expression of the anti-apoptotic protein BCL-2 under the influence of the combined action of beetroot extract and doxorubicin, which in turn increased the BAX:BCL-2 in the MDA-MB-231 cell line, making the cells more sensitive to this mixture. In studies on colorectal cancer cells, Saber et al. [44] showed that both red beetroot extract and betanin induced apoptosis pathways (internal and external) by reducing the expression of the anti-apoptotic gene *BCL-2* and increasing the expression of the pro-apoptotic gene *BAD*.

As a result of apoptosis, cytochrome c is released, which is generally located on the inner side of the mitochondrial membrane and acts as an electron transmitter; Smack/DIABLO and HtrA2/Omi proteins are also transferred from the mitochondria to the cytoplasm. An increase in the expression of the proteins mentioned above, confirmed in our own research using the Western blot technique, was observed in the MCF-7 (Figure 3) and partly in the MDA-MB-231 cell lines (in the case of HtrA2/Omi, there was a decrease) (Figure 4).

Release of cytochrome c, lowering the potential of the mitochondrial membrane along with the activation of the intrinsic apoptotic pathway. after exposure to betanin, was demonstrated by Sreekanth et al. [41], but in studies with human cells (chronic myelogenous leukemia cells of the K562 lineage).

In the cytoplasm, cytochrome c interacts with the Apaf1 protein and procaspase-9, forming the so-called apoptosome [53] that activates procaspase-9 [55], followed by effector caspases, including caspase-3 and -7 [56].

The present study showed a significant increase in the expression of the *APAF1* gene in MCF-7 (Appendix A) and MDA-MB-231 (Appendix A) cell lines under the influence of digested juice from young shoots, and a tendency for increased or a significant increase in this gene expression in MCF-7 (Appendix A) and MDA-MB-231 (Appendix A) cell lines, treated with digested root juice. Although the results presented in the paper clearly indicate the induction of apoptotic events as a result of the action of digested juice from young beetroot shoots, the expression of the *CASP3* gene was not detected in the MCF-7 line, which may be explained by the presence of a functional deletion [51]. In addition, some studies have indicated that apoptosis can proceed without the activation of caspase-3 but with caspase-7 [52]. The presented results confirm this finding because after treating the cells with the analyzed material (both digested juice of young shoots and the root), a tendency was shown for the enhanced expression *CASP7* gene in the MCF-7 cell line (Appendix A).

The gene expression results correlated with increased protein expression in the cells of the estrogen-dependent line (Figure 3). The second analyzed cell line showed a significantly increased expression of *CASP3* and partially caspase-7 (no significant changes in the case of digested root juice) at the level of active protein (Figure 4).

In the study with the human lung cancer A549 cell line, Zhang et al. [57] showed that betanin induced apoptosis by activating caspases-3, -7, and -9. In the research with colorectal cancer cells of the HT-29 and Caco-2 lines, Saber et al. [44] found that the combined action of beetroot extract and betanin resulted in the induction of apoptosis pathways, among others, by increasing *CASP-3* and *CASP-9* gene expression.

Poly(ADP-ribose) polymerase (PARP) is a nuclear enzyme essential for maintaining genome integrity. During apoptosis, it is one of the first proteins to be degraded by the action of caspase-3 and -7 in order to prevent cell survival. Cleavage of the PARP protein and the occurrence of its fragments (the so-called cut form) of low molecular weight is a characteristic marker of the apoptosis process [58].

In the present study, a significantly increased activity of the cleaved form of this protein was demonstrated in the MCF-7 cancer cell line under the influence of both types of juice (Figure 3). In turn, the MDA-MB-231 cell line showed a significant increase in PARP cut as a result of the action of only digested juice from young shoots (Figure 4).

Zhang et al. [57] showed that betanin induced apoptosis by increasing the expression of the PARP protein in the lung cancer A549 cell line. Das et al. [39] also showed that a mixture of beetroot root extract and doxorubicin significantly increased the expression of the cleaved form of the PARP protein.

Another pro-apoptotic protein released from mitochondria during apoptosis is AIF, which moves to the nucleus and causes DNA fragmentation and chromatin condensation [59]. The results obtained in this study showed a significant increase in mRNA expression of the gene encoding the AIFM1 protein in cancer cells of both lines as a result of the action of digested juice from young shoots (Appendix A). In turn, a significant increase in the expression of this gene under the influence of digested root juice was shown only in the estrogen-independent cells (Appendix A).

Two different pathways leading to the process of programmed death, that is, intrinsic via mitochondrial proteins and extrinsic via death receptors, are often characterized by common elements such as activation of effector caspases and release of the Smac/DIABLO group of proteins from the mitochondrion during an apoptotic signal derived from the receptor [60].

The p53 protein, activated after cell DNA damage, also affects the mitochondrion, releasing the BAX protein, and affects the transcription of the Fas ligand gene, stimulating its production [61].

During the binding of the ligand to the membrane death receptor, caspase-8 may also be activated and, as a result, trigger the external apoptosis pathway. The initiating caspase-8 is responsible for forming the DISC complex and activating procaspase-3, -6 and -7. Active caspase-8 cleaves the BID protein, creating the tBID protein, which, by increasing the permeability of the mitochondrial membrane, releases cytochrome c and, consequently, links receptor-based and mitochondrial apoptosis.

In our own research, a significantly higher expression of the *CASP8* gene was demonstrated in the MCF-7 cell line under the influence of digested juice from young shoots and under the influence of both types of juices in the MDA-MB-231 cell line (Table 7 and Appendix A). The increase in *CASP8* gene expression correlated with the increase in the level of caspase-8, assessed by Western blot (Figure 3 and Figure 4).

Scarpa et al. [43] also showed an increased expression of caspase-8 under the influence of betalains, except in bladder cancer cells of the T24 line. On the other hand, in studies with colon cancer cells of the HT-29 and Caco2 lines, Saber et al. [44] showed that a mixture of beetroot extract with betanin enhanced the expression of the *CASP8* gene.

In the MDA-MB-231 cell line, a significantly reduced expression of the *FADD* gene was demonstrated due to the action of both types of digested juice analyzed (Table 7 and Appendix A). In addition, the MCF-7 cell line also showed reduced expression of the *Fas* gene when exposed to the digested juice from young shoots, while no changes were observed in the cells treated with digested root juice (Table 7 and Appendix A).

Opposite results were obtained by Nowacki et al. [38], who observed the increased expression of the *Fas* protein under the influence of beetroot extract enriched in betanin in the MCF-7 cell line. In turn, in studies with colon cancer cells of the CaCo-2 line, Saber et al. [44] showed that red beetroot extract, in combination with betanin, increased the expression of the *Fas* gene, inducing the extrinsic apoptosis pathway.

In the present study, a significant reduction in the expression of RIP protein was additionally demonstrated based on Western blot analysis in both analyzed research models (Figure 3 and Figure 4) under the influence of both types of digested juice. RIP is essential for TNF-α-induced activation of NF-κB. This study showed a significant reduction in *NF-κB* gene expression in the MCF-7 cell line under the influence of digested juice from young shoots of beetroot and no effect of digested root juice (Table 7 and Appendix A). Interestingly, in the second analyzed line, a significant increase in the expression of this gene was demonstrated under the influence of both discussed factors (Table 7 and Appendix A).

NFκB enters the cell nucleus and initiates anti-apoptotic gene transcription that promotes survival. This is consistent with the results of our own work described above. In most cases, lower effectiveness of reducing the proliferation of MDA-MB-231 breast cancer cell line was demonstrated compared with the MCF-7 line, under the influence of the tested material.

In this work, the impact of the analyzed material on the expression of selected proteins was observed in several cellular processes, regulating proliferation, differentiation and the cell cycle. One of the most important pathways of signaling in the cell, involved in the process of oncogenesis, is the AKT kinase pathway. Although activation of AKT kinase alone is not believed to be sufficient for the onset of cancer, it plays an essential role in its progression by preventing apoptosis, promoting changes in the metabolism of the diseased cell, and regulating processes related to cell proliferation and migration.

The *AKT1* gene encodes a protein with serine/threonine kinase activity and is involved in signaling pathways related to survival and proliferation.

In turn, the *PHLPP1* gene encodes a phosphatase involved in the dephosphorylation of the AKT1 kinase.

In this work, it has been shown that the expression of the *AKT1* gene, responsible for proliferation enhancement due to the activation of the PI3K/Akt pathway, was significantly reduced in the cells of both analyzed lines under the influence of digested juices from both young shoots and the root (Table 7, Appendix A).

Another kinase evaluated in this study was p38 mitogen-activated kinase. The study showed that the juice from young shoots and beetroot root, subjected to digestion and absorption, significantly increased the expression of p38 MAPK in the MDA-MB-231 cell line (Figure 4). On the other hand, a significant increase in the expression of the discussed protein in the MCF-7 estrogen-dependent breast cancer cell line was demonstrated as a result of the action of digested root juice, while no significant changes were observed as a result of cell treatment with digested juice from young shoots (Figure 3).

According to Swat et al. [62], mitogen-activated kinase p38 can inhibit cell proliferation by antagonizing the JNK/c-Jun pathway, involved in the regulation of proliferation and apoptosis. One of the most important functions of the p38 protein is the promotion of cell differentiation. Research also shows that the activation of the p38 kinase pathway in different types of tumors may be downregulated. Genetic modifications of the proteins involved in this pathway showed that p38 may act as a tumor suppressor. Research shows that p38 can also inhibit cell proliferation by modulating the expression of epithelial growth factor receptors (EGFR) [63]. Overexpression or mutation of EGFR receptors in cancer cells may lead to dysregulation of the mechanism of signaling through these receptors, resulting in excessive proliferation, increased angiogenesis, and, consequently, migration of cancer cells to the surrounding tissues. This information is crucial in the light of the research results presented in this paper, according to which the analyzed juice modulated the expression level of p38 kinase in the MDA-MB-231 cell line, characterized by overexpression of the EGFR receptor.

The above-described effect of the juice from young shoots of beetroot, subjected to digestion and absorption, is also important in activation of the p53 transcription factor by the p38 mitogen-activated kinase, and consequently inhibition of cell proliferation.

Another gene analyzed in this work was *MYC*, encoding a protein C-MYC, which is also involved in regulating proliferation and apoptosis. Our study results showed that digested juice from both young shoots and beetroot root significantly increased the *MYC* gene expression in the cells of both analyzed lines (Table 7, Appendix A).

The described mechanism of action of digested juice from young shoots in the MCF-7 breast cancer cell line indicates the involvement of the internal apoptosis pathway with the participation of executive caspases and the p53 transcription factor. The second signaling pathway whose activity changed under the influence of the analyzed juice was the pathway in which AKT proteins are involved. Reducing the activity of these proteins resulted in the inhibition of cell proliferation and survival. In turn, in the MDA-MB-231 cell line, MAP kinases and the NF-κB factor were additionally activated. The increase observed in the expression of caspase-8 in the cells of both research models, with the simultaneous lack of an increase in the expression level of the analyzed death receptors, requires further research.

In turn, by analyzing the results of own research on the impact of digested beet root juice, it can be concluded that similar mechanisms were activated in MCF-7 and MDA-MB-231 cell lines, resulting in the release of cytochrome c and activation of the Smak/DIABLO protein (only in the MCF-7 line), caspases-7, -8 and PARP, but to a lesser extent compared to the effect of digested juice from young shoots.

## 4. Materials and Methods

### 4.1. Plant Material

The analyzed material included ready-to-eat, fourteen-day young shoots and roots at full maturity of red beet (*Beta vulgaris* L., Polish cultivar Rywal). The young shoots were grown in sowing boxes filled with a standard garden substrate in late May. The same seedlings were planted into the ground. Beetroots were sown on brown soil with pH 6.5, salinity 0.57 g/L NaCl, NH_4_ 3.5 mg/L, NO_3_ 52.5 mg/L, P 187 mg/L, K 187 mg/L, Ca 1324 mg/L, and Mg 188.45 mg/L. The young shoots were analyzed in their entirety. Samples of mature vegetables were cleaned of soil, washed, and prepared by cutting different-sized roots. The fresh material was used to prepare juices (by squeezing in a juicer MPM). After squeezing, the samples were enclosed in 10-mL plastic falcons and stored in a freezer (−20 °C) until analyzed. The material was thawed prior to analysis.

### 4.2. Determination of Polyphenolic Profile

The HPLC analysis of polyphenols was conducted using the Prominence-i LC-2030C 3D Plus system (Shimadzu, Kyoto, Japan) equipped with a diode array detector (DAD) according to Dziadek et al. [64]. The separation was performed on the Luna Omega 5 μm Polar C18, 100 A, 250 mm × 10 mm column (Phenomenex, CA, USA) at 40 °C. The mobile phase was a mixture of two eluents: A—0.1% formic acid in water (*v*/*v*) and B—0.1% formic acid in methanol (*v*/*v*). The flow rate of the mobile phase was 1.2 mL min^−1^. The analysis was carried out with the following gradient conditions: from 20% to 40% B in 10 min, 40% B for 10 min, from 40% to 50% B in 10 min, from 50% to 60% B in 5 min, 60% B for 5 min, from 60 to 70% B in 5 min, from 70% to 90% B in 5 min, 90% B for 5 min, from 90% to 20% B (the initial condition) in 1 min and 20% B for 4 min, resulting in a total run time of 60 min. The injection volume was 20 μL. The detection of 4-hydroxybenzoic acid, myricetin, isorhamnetin, quercetin, and luteolin was performed at 254 nm; rutin at 256 nm; vanillic acid at 260 nm; kaempferol at 264 nm; apigenin at 267 nm; gallic acid at 271 nm; hispidulin at 273 nm; syringic acid at 274 nm; catechin and epicatechin at 278 nm; naringin and carnosol at 283 nm; hesperidin and carnosic acid at 284 nm; *p*-coumaric acid at 310 nm; caffeic acid, ferulic acid and sinapinic acid at 323 nm; chlorogenic acid at 326 nm; and rosmarinic acid at 329 nm. Quantitative determinations were carried out using calibration curves of the standards. Individual stock standard solutions of each phenolic compound were prepared in 0.1% formic acid in 70% methanol at a concentration of 100 mg L^−1^. The calibration curves were made by using the working standard solutions, which were prepared by mixing suitable volumes of each stock standard solution, to give concentrations ranging from 0.5 to 4.0 mg L^−1^. All working standard solutions and studied samples were filtered through a 0.22 μm pore size membrane filter. The data was integrated and analyzed using the LabSolutions software ver. 5.93 (Shimadzu Corporation, Kyoto, Japan).

### 4.3. Simulated In Vitro Digestion Model of the Gastrointestinal Tract

Digestion of the juice made of young shoots and mature root of beetroot was performed in vitro in two steps, according to Minekus et al. [65]. Pepsin (Sigma-Aldrich, St. Louis, MO, USA) was used and pH was adjusted to pH = 2.0 (the mixture was incubated at 37 °C for 2 h, with shaking) to create conditions reflecting the in vivo processes occurring in the stomach. To make the environment similar to that of the intestine, pancreatin from porcine pancreas (Sigma-Aldrich, St. Louis, MO, USA) and bile salts (Sigma-Aldrich, St. Louis, MO, USA) were added, and pH was increased to 7.4 (the mixture was incubated at 37 °C for 2 h, with shaking). The next step involved the introduction of intestinal microflora and incubation at 37 °C for 16 h.

### 4.4. Simulated In Vitro Absorption Model of the Gastrointestinal Tract

The digested in vitro product was sterilized by filtration through nitrocellulose filters. A human colon cancer cell line Caco-2 was used to study the transport of digestive products. The culture of Caco-2 cells leads to the establishment of monolayers with intermediate properties regarding trans-epithelial electrical resistance (TEER). Cells were harvested at 90% confluence with trypsin-EDTA and seeded onto PET inserts (BIOKOM, Janki, Poland), mounted in 12-well plates, at a total density of 1 × 10^5^ cells per well. The culture medium was replaced every other day, and the cell’s monolayer was grown for 21 days. Caco-2 monolayers with an integrity equivalent to a trans-epithelial electrical resistance (TEER) higher than 200 Ω/cm^2^ were used for permeability experiments. The Caco-2 monolayer was washed twice with PBS solution (Sigma-Aldrich, ST. Louis, MO, USA). Then, the digested juices were placed on the top (apical side) of the Caco-2 cells monolayer. After 2 h of incubation, the filtrate was collected and secured for subsequent analysis.

### 4.5. Cell Culture

The human breast adenocarcinoma MCF-7 (hormone-dependent HTB-22TM) and MDA-MB-231 cell lines (hormone-independent HTB-26™), the human colon cancer cell line Caco-2 (HTB-37™), and the human normal breast MCF-12A (CRL-10782) cell line were purchased from the American Type Culture Collections (ATCC, Manassas, VA, USA). Cells were cultured in the appropriate medium, with fetal bovine serum (FBS), supplements and antibiotic mixture (Sigma-Aldrich, St. Louis, MO, USA) according to the ATCC^®^ protocol. The MCF-7, MCF-12A, and Caco-2 cell lines were stored in an incubator (NuAire, Plymouth, MN, USA) at 37 °C in a 5% CO_2_ atmosphere. The MDA-MB-231 cell line was stored in an incubator (Thermo Fisher Scientific, Waltham, MA, USA) at 37 °C in 100% air. The culture medium was replaced with fresh medium every 2–3 days, and passage was performed when 80% confluence was reached.

### 4.6. Cell Treatment

The MCF-7 and MDA-MB-231 cells were seeded on 96-well sterile plates at 8 × 10^4^ cells per well, with 12-well plates at 9 × 10^4^ cells per well, 6-well plates at 2 × 10^5^ cells per well or Petri dishes, depending on the experiment. Twenty-four hours after seeding, the culture medium was replaced with (1) medium with the addition of juice from young shoots of beetroot in its native form in the proportions 9:1, 8:2, 7:3, 6:4 and 5:5 (*v*/*v*); (2) medium with the addition of juice from root of beetroot in its native form in the proportions as above, (3) juice from young beetroot shoots subjected to in vitro gastrointestinal digestion and absorption process, and (4) beetroot root juice subjected to the process as above for 24, 48 and 72 h. Untreated cells (UC) grown in the complete culture medium, without any juices, were used as a negative control of all experiments. Triton X-100 [2%] (Sigma-Aldrich, St. Louis, MO, USA) was used as a positive control to solubilize membrane proteins in the LDH assay. Staurosporine [1.5 µM for 3 h] (Sigma-Aldrich, St. Louis, MO, USA) was used as a positive control of the activation of apoptotic proteins, the process of apoptosis, and as a confirmation of the correctness of the assumed experimental conditions.

### 4.7. Cell Proliferation Assessment

Cell proliferation was determined using a commercial colorimetric and immunoassay Cell Proliferation ELISA kit, BrdU (Roche, Basel, Switzerland), based on the measurement of BrdU incorporation during DNA synthesis. The assay was carried out according to the manufacturer’s protocol. All results were standardized to the untreated control (UC) as 100%.

### 4.8. Cytotoxicity Assay

The cytotoxicity assessment of the material was performed using the Cytotoxicity Detection Kit (LDH) (Roche, Basel, Switzerland) according to the protocol provided by the manufacturer.

### 4.9. The Muse^®^ Flow Cytometer Analysis

The cells were labelled according to the protocol provided by the manufacturer for the analyses: Muse^®^ Annexin V & Dead Cell Assay Kit, Muse^®^ Bcl-2 Activation Dual Detection Assay, and Muse^®^ MultiCaspase Assay Kit. Analyses were performed using a Muse^®^ Cell Analyzer (Merck, Kenilworth, NJ, USA).

### 4.10. RNA Isolation, cDNA Synthesis, and RT-qPCR

The total RNA was isolated from cancer cell cultures using a Total RNA Mini Plus Kit (A&A Biotechnology, Gdynia, Poland), following the manufacturer’s instructions. The concentration, purity, and quality of isolated RNA were measured in µDrop Plate (Thermo Fisher Scientific, Waltham, MA, USA). The reverse transcriptase (RT) reaction was performed to synthesize cDNA using iScript Reverse Transcription Supermix (Bio-Rad, Hercules, CA, USA). Quantitative verification of gene expression was performed using the CFX96 TouchTM Real-Time PCR Detection System instrument (Bio-Rad, Hercules, CA, USA). The PCR reaction mixture contained cDNA samples, RNase-free water, SYBR Green Supermix (Bio-Rad, Hercules, CA, USA), and primers for the following genes: apoptosis-inducing factor mitochondria associated 1 (AIFM1), AKT serine/threonine kinase 1 (AKT1), apoptotic peptidase activating factor 1 (Apaf-1), Bcl2-associated agonist of cell death (BAD), Bcl2- binding component 3 (BBC3), B Cell Lymphoma 2 (BCL2), H3-interacting domain death agonist (BID), caspase-3 (CASP-3), caspase-7 (CASP-7), caspase-8 (CASP-8), diablo IAP-binding mitochondrial protein (DIABLO), Fas-associated death domain (FADD), Fas cell surface death receptor (FAS), v-myc myelocytomatosis viral oncogene homolog (MYC), nuclear factor of kappa light polypeptide gene enhancer in B-cells 1 (NFKB1), PH domain and leucine-rich repeat protein phosphatase 1 (PHLPP1) and tumor protein p53 (TP53). The conditions of individual PCR reactions were optimized for the given pair of oligonucleotide primers. Amplification was performed using the following conditions: activation at 95 °C for 2 min, 40 cycles of denaturation at 95 °C for 5 sec, annealing at 60 °C for 30 sec and melt curve at 65–95 °C for 5 sec. Results were normalized using reference gene β-actin (ACTB) and were determined using the 2−∆∆CT method as previously reported by Livak and Schmittgen [66].

### 4.11. Western Blot Assays

Cell lysis was carried out with use of Cell Lysis Buffer (Cell Signaling Technology, Danvers, MA, USA) with the addition of Protease Inhibitor Cocktail (BioShop, Burlington, ON, Canada) according to the protocol provided by the manufacturer. The amount of protein in the cell lysates was determined using the Pierce™ BCA Protein Assay Kit (Thermo Fisher Scientific, Waltham, MA, USA) and the Multiscan GO Microplate Reader (Thermo Fisher Scientific, Waltham, MA, USA). To examine the mechanism of action of the experimental juices, cell lysate proteins were separated using Mini-PROTEAN TGX Stain-Free precast gels (Bio-Rad, Hercules, CA, USA). The separation was conducted in electrophoresis buffer (3.03 g Trizma Base, 14.4 g glycine, 1 g SDS, made up to 1 L with water), for 70 min, at 100 V. Next, the Trans- Turbo Mini Nitrocellulose Transfer Packs (Bio-Rad, Hercules, CA, USA) were used, for the fast, efficient transfer of proteins from gels using the Trans-Blot^®^ TurboTM Transfer System (Bio-Rad, Hercules, CA, USA). Subsequently, the immobilized proteins were incubated with the appropriate primary antibody: cytochrome c (#11940), caspase-7 (#9492), caspase-8 (#4790), mitochondrial serine protease HtrA2/Omi (#9745), Poly (ADP-ribose) Polymerase PARP (#9542), a second mitochondria-derived activator of caspase (Smak/Diablo) (#2954), receptor-interacting protein kinase RIP (#3493), p38 MAPK (D13E1) tumor protein p53 (#2527), and β-tubulin (#4970) as a reference protein (Cell Signaling Technology). Finally, the appropriate secondary antibody conjugated with horseradish peroxidase #7076, Cell Signaling Technology) was applied. Detection was executed by chemiluminescence, using ClarityTM Western ECL Substrate (Bio-Rad, Hercules, CA, USA). The detected protein was visualized using the ChemiDocTM Imaging System (Bio-Rad, Hercules, CA, USA). Densitometric assays were performed using ImageJ (author Wayne Rasband). Results are shown as a mean ± SD normalized to the internal reference protein (β-tubulin). Untreated negative control (UC) was set as 100% expression level.

### 4.12. Statistical Analysis

The statistical analysis was performed using Statistica 13.1 PL software (StatSoft, Inc., Tulsa, OK, USA). All experiments were performed in at least three technical and biological replications. Results are expressed as mean ± standard deviation (SD).

## 5. Conclusions

The present study results provide evidence for the apoptotic effect of the evaluated types of beetroot juice, in particular those made of young shoots or the root and subjected to the process of digestion and absorption in a model gastrointestinal tract in vitro, against cells of both analyzed breast cancer lines.

In conclusion, it can be stated that research into the anti-carcinogenic activity of young shoots is considered to be at an early stage. Therefore, a continuation of research in this area is justified. In addition, the previously conducted process of digestion and absorption makes it possible to consider both the bioavailability of the compounds contained in the product and their combined (synergistic/antagonistic) effect on the body’s cells.

## Figures and Tables

**Figure 1 ijms-24-06889-f001:**
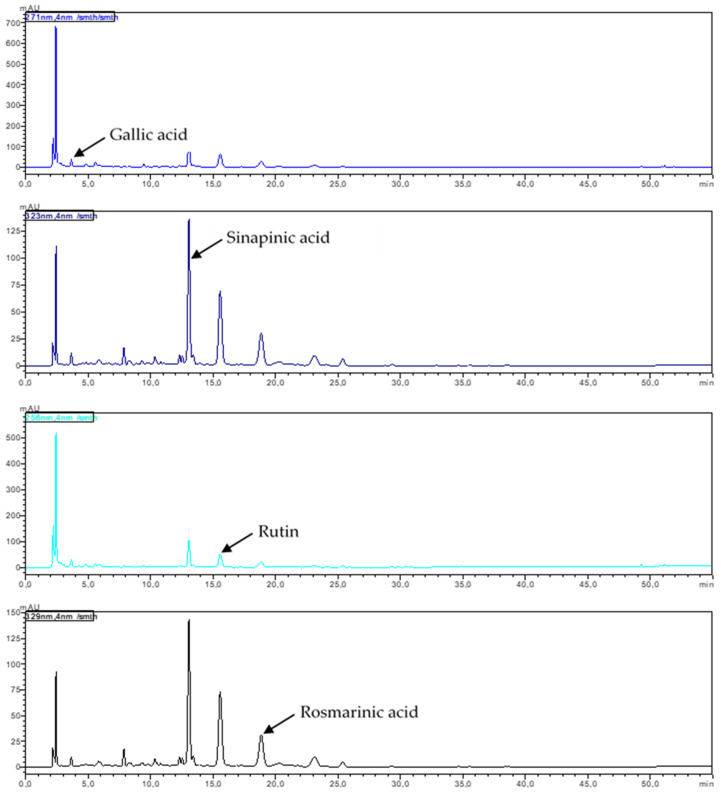
HPLC-DAD chromatograms of selected polyphenols identified in the juice from young shoots of beetroot.

**Figure 2 ijms-24-06889-f002:**
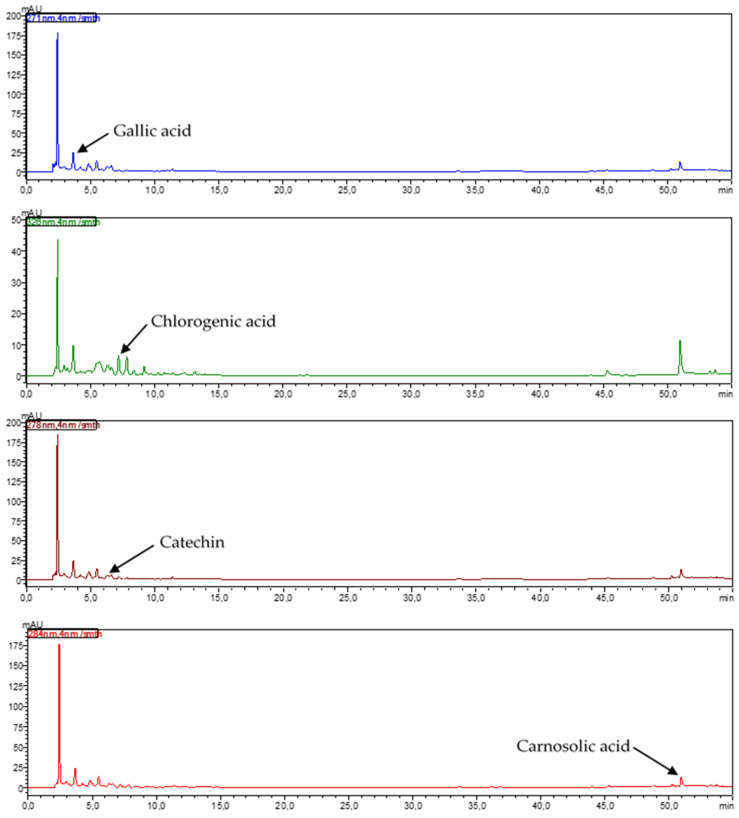
HPLC-DAD chromatograms of selected polyphenols identified in the juice from the root of beetroot at full maturity.

**Figure 3 ijms-24-06889-f003:**
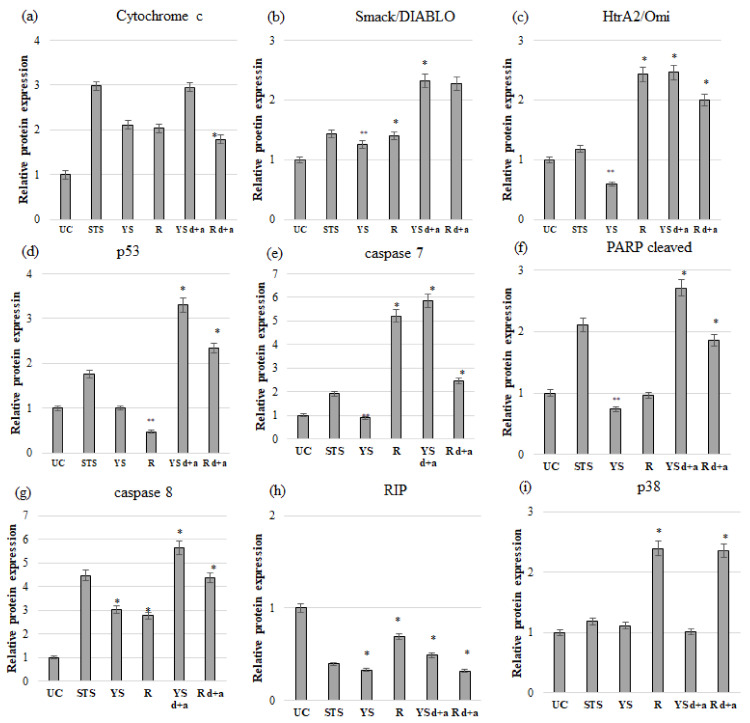
Expression of selected apoptosis-related proteins in MCF-7 cancer cell line. UC—untreated control, STS—staurosporine, YS—native juice of young shoots of beetroot, R—native juice of beetroot, YS d+a—juice of young shoots of red beet subjected to in vitro gastrointestinal digestion and absorption, R d+a—juice of beetroot subjected to in vitro gastrointestinal digestion and absorption, (**a**–**i**) analyzed proteins. * *p* ≤ 0.05 compared to controls, U Mann–Whitney test; ** *p* ≤ 0.1 compared to controls, U Mann–Whitney test.

**Figure 4 ijms-24-06889-f004:**
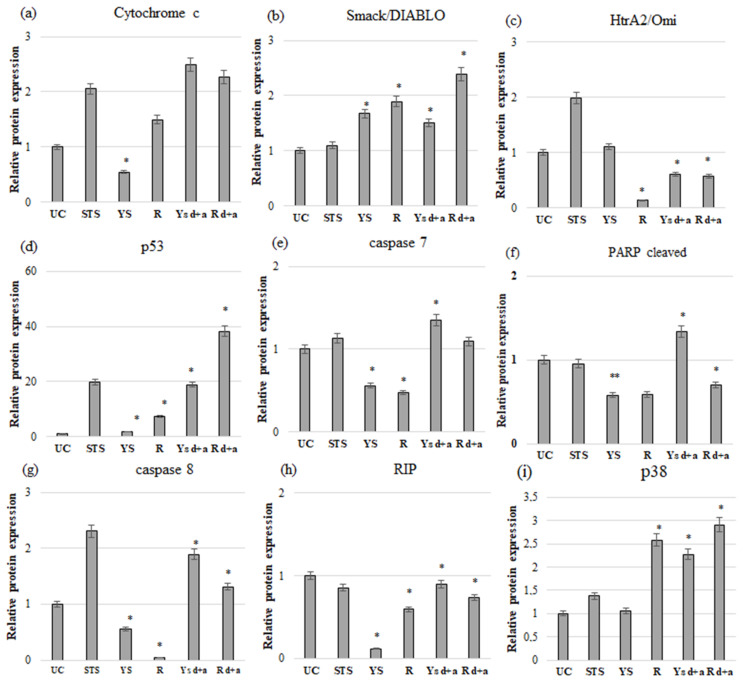
Expression of selected apoptosis-related proteins in MDA-MB-231 cancer cell line. UC—untreated control, STS—staurosporine, YS—native juice of young shoots of beetroot, R—native juice of beetroot, YS d+a—juice of young shoots of red beet subjected to in vitro gastrointestinal digestion and absorption, R d+a—juice of beetroot subjected to in vitro gastrointestinal digestion and absorption, (**a**–**i**) analyzed proteins. * *p* ≤ 0.05 compared to controls, U Mann–Whitney test; ** *p* ≤ 0.1 compared to controls, U Mann–Whitney test.

**Table 1 ijms-24-06889-t001:** Comparison of the contents of polyphenolic compounds in juices made from young shoots and the root of beetroot at full maturity [mg/100 mL of juices].

Polyphenolic Compounds	Retention Time [min]	Part of a Beetroot	
		Young Shoots	Root
		Mean ± SD	Mean ± SD
Gallic acid	3.655	28.08 ^a^ ± 0.02	5.09 ^b^ ± 0.02
Chlorogenic acid	7.171	1.86 ^a^ ± 0.01	1.42 ^b^ ± 0.03
4-Hydroxybenzoic acid	7.870	4.23 ^a^ ± 0.01	0.30 ^b^ ± 0.02
Caffeic acid	8.873	1.24 ^a^ ± 0.00	0.11 ^b^ ± 0.00
Vanillic acid	9.059	1.28 ^a^ ± 0.04	0.11 ^b^ ± 0.01
Syringic acid	9.794	1.95 ^a^ ± 0.01	0.09 ^b^ ± 0.01
*p*-Coumaric acid	11.773	0.61 ^a^ ± 0.00	0.20 ^b^ ± 0.01
Ferulic acid	12.537	4.38 ^a^ ± 0.00	0.12 ^b^ ± 0.01
Sinapinic acid	13.065	222.81 ^a^ ± 0.25	0.06 ^b^ ± 0.00
Rosmarinic acid	18.870	171.56 ± 0.01	Nd
Catechin	6.252	27.75 ^a^ ± 0.17	9.33 ^b^ ± 0.17
Epicatechin	8.253	29.84 ^a^ ± 0.01	2.22 ^b^ ± 0.16
Naringin	14.080	Nd	0.35 ± 0.02
Hesperidin	15.157	190.74 ^a^ ± 0.02	0.30 ^b^ ± 0.02
Rutin	15.566	172.31 ^a^ ± 0.07	0.11 ^b^ ± 0.01
Myricetin	17.235	12.03 ± 0.01	Nd
Quercetin	25.374	13.22 ± 0.20	Nd
Luteolin	29.320	2.19 ± 0.02	Nd
Hispidulin	33.670	2.07 ± 0.00	Nd
Apigenin	34.929	0.50 ± 0.00	Nd
Kaempferol	35.954	3.67 ± 0.00	Nd
Isorhamnetin	37.046	4.02 ± 0.01	Nd
Carnosol	49.299	16.47 ^a^ ± 0.04	0.53 ^b^ ± 0.01
Carnosolic acid	50.982	10.73 ^a^ ± 0.71	2.17 ^b^ ± 0.04
Total		923.54 ^a^	22.51 ^b^

^a, b^—values marked in rows with different letters differ statistically significantly (*p* ≤ 0.05), Nd—not detected.

**Table 2 ijms-24-06889-t002:** Comparison of changes in the proliferation of MCF-7 and MDA-MB-231 breast cancer cell lines (% control).

MCF-7 Cell Line						
Incubation Time [h]	24		48		72	
Type of Juice	YS	R	YS	R	YS	R
Juice:Medium (*v*/*v*)	Mean ± SD					
1:9	106.99 ^A^ ± 0.02	109.22 ^A^ ± 0.08	117.31 ^aA^ ± 0.05	124.14 ^bA^ ± 0.05	115.82 ^aA^ ± 0.03	119.67 ^bA^ ± 0.07
2:8	98.20 ^A^ ± 0.07	100.45 ^A^ ± 0.23	108.03 ^aA^ ± 0.20	116.66 ^bA^ ± 0.06	106.13 ^aA^ ± 0.15	101.11 ^bA^ ± 0.03
3:7	92.00 ^aA^ ± 0.10	97.70 ^bA^ ± 0.03	89.59 ^a^ ± 0.05	93.27 ^b^ ± 0.08	79.27 ^aA^ ± 0.08	98.36 ^bA^ ± 0.07
4:6	87.93 ^A^ ± 0.08	90.94 ^A^ ± 0.15	74.60 ^A^ ± 0.03	77.51 ^A^ ± 0.07	52.50 ^aA^ ± 0.08	64.96 ^bA^ ± 0.03
5:5	85.01 ^aA^ ± 0.01	88.97 ^bA^ ± 0.37	59.52 ^A^ ± 0.01	58.34 ^A^ ± 0.02	48.23 ^aA^ ± 0.01	57.45 ^bA^ ± 0.02
Digested and absorbed juice	72.62 ^aA^ ± 0.05	77.03 ^bA^ ± 0.04	66.96 ^A^ ± 0.02	69.41 ^A^ ± 0.03	41.17 ^aA^ ± 0.01	36.54 ^bA^ ± 0.03
MDA-MB-231 cell line						
1:9	121.04 ^aB^ ± 0.03	113.63 ^bB^ ± 0.03	100.03 ^aB^ ± 0.01	116.32 ^bB^ ± 0.07	98.11 ^aB^ ± 0.02	106.96 ^bB^ ± 0.04
2:8	134.79 ^aB^ ± 0.01	123.30 ^bB^ ± 0.01	95.28 ^aB^ ± 0.01	101.20 ^bB^ ± 0.04	94.19 ^B^ ± 0.02	96.77 ^B^ ± 0.07
3:7	97.28 ^B^ ± 0.01	99.40 ^B^ ± 0.01	88.50 ^a^ ± 0.05	92.19 ^b^ ± 0.08	90.56 ^aB^ ± 0.03	87.15 ^bB^ ± 0.06
4:6	92.97 ^aB^ ± 0.01	96.33 ^bB^ ± 0.01	85.63 ^aB^ ± 0.03	90.04 ^bB^ ± 0.08	85.84 ^B^ ± 0.07	83.11 ^B^ ± 0.07
5:5	89.93 ^aB^ ± 0.02	95.14 ^bB^ ± 0.03	84.28 ^B^ ± 0.07	86.82 ^B^ ± 0.07	80.18 ^aB^ ± 0.03	75.93 ^bB^ ± 0.07
Digested and absorbed juice	95.24 ^aB^ ± 0.02	104.30 ^bB^ ± 0.02	90.89 ^B^ ± 0.05	93.08 ^B^ ± 0.08	86.81 ^aB^ ± 0.04	90.19 ^bB^ ± 0.01

YS—native juice from young shoots of beetroot, R—native juice from root of beetroot, Values are expressed as means ±SD for n = 9, standardized to untreated control set as 100%. ^a, b^—values marked in rows with different letters differ statistically significantly (*p* ≤ 0.05) for every three pairs of columns (YS vs. R) in the same time interval. ^A, B^—values marked in columns with different letters differ statistically significantly (*p* ≤ 0.05) between both analyzed cell lines, for the same juice-to-medium ratio, in analogous time intervals.

**Table 3 ijms-24-06889-t003:** Comparison of the cytotoxicity of the analyzed juices against MCF-7 and MDA-MB-231 breast cancer cell lines [%].

MCF-7 Cell Line						
Incubation Time [h]	24		48		72	
Type of Juice	YS	R	YS	R	YS	R
Juice:Medium (*v*/*v*)			Mean ± SD			
1:9	2.01 ± 0.01	1.10 ± 0.01	1.41 ± 0.04	0.30 ± 0.01	2.07 ± 0.03	2.23 ± 0.07
2:8	5.04 ^A^ ± 0.02	5.50 ^A^ ± 0.04	2.26 ± 0.02	2.26 ± 0.02	0.55 ^a^ ± 0.01	4.92 ^bA^ ± 0.02
3:7	8.77 ± 0.01	7.27 ^A^ ± 0.01	8.58 ^A^ ± 0.01	6.95 ^A^ ± 0.01	9.92 ^A^ ± 0.01	8.11 ± 0.04
4:6	17.37 ^A^ ± 0.03	18.34 ^A^ ± 0.01	23.00 ^aA^ ± 0.01	13.98 ^bA^ ± 0.02	21.31 ^A^ ± 0.01	20.18 ^A^ ± 0.02
5:5	23.97 ^A^ ± 0.02	21.41 ^A^ ± 0.13	30.93 ^aA^ ± 0.04	24.50 ^bA^ ± 0.03	35.47 ^A^ ± 0.03	33.63 ^A^ ± 0.05
Digested and absorbed juice	3.23 ^a^ ± 0.01	6.52 ^bA^ ± 0.12	9.95 ± 0.07	8.27 ^A^ ± 0.01	12.31 ± 0.04	17.34 ^bA^ ± 0.06
MDA-MB-231 cell line						
1:9	0.39 ± 0.01	0.77 ± 0.01	0.17 ± 0.01	0.25 ± 0.02	0.33 ± 0.01	0.20 ± 0.01
2:8	0.67 ^B^ ± 0.02	0.23 ^B^ ± 0.02	1.01 ± 0.01	0.33 ± 0.01	1.49 ± 0.01	1.10 ^B^ ± 0.01
3:7	1.59 ± 0.04	0.68 ^B^ ± 0.02	5.17 ^aB^ ± 0.06	3.52 ^bB^ ± 0.02	5.35 ^B^ ± 0.01	6.45 ± 0.03
4:6	5.79 ^aB^ ± 0.01	2.06 ^bB^ ± 0.04	6.96 ^B^ ± 0.03	8.34 ^B^ ± 0.02	7.51 ^B^ ± 0.03	9.57 ^B^ ± 0.04
5:5	8.38 ^aB^ ± 0.01	1.74 ^bB^ ± 0.01	7.28 ^B^ ± 0.08	8.66 ^B^ ± 0.02	10.52 ^B^ ± 0.09	11.04 ^B^ ± 0.04
Digested and absorbed juice	4.22 ^a^ ± 0.03	1.32 ^bB^ ± 0.03	8.25 ^a^ ± 0.07	4.66 ^bB^ ± 0.04	9.78 ^a^ ± 0.04	6.35 ^bB^ ± 0.05

YS—native juice from young shoots of beetroot, R—native juice from root of beetroot. ^a, b^—values marked in rows with different letters differ statistically significantly (*p* ≤ 0.05) for every three pairs of columns (YS vs. R) over the same time frame. ^A, B^—values marked in columns with different letters differ statistically significantly (*p* ≤ 0.05) between both analyzed cell lines, for the same juice concentrations, in analogous time intervals.

**Table 4 ijms-24-06889-t004:** Comparison of apoptosis induction in cancer cells [%] using flow cytometry (Muse^®^ Cell Analyzer).

MCF-7 Cell Line						
Cells	UC	STS	YS	R	YS d+a	R d+a
	Mean ± SD					
Live	76.80 ± 0.52	38.13 ± 0.14	68.53 ^aA^ ± 1.04	52.53 ^b^ ± 3.00	41.93 ^A^ ± 3.54	44.20 ^A^ ± 2.59
Total apoptotic	22.93 ± 0.66	61.27 ± 0.14	29.67 ^aA^ ± 1.41	47.40 ^b^ ± 7.69	56.67 ^A^ ± 2.93	55.50 ^A^ ± 2.64
Early Apoptotic	19.93 ± 1.98	44.27 ± 0.37	17.67 ^A^ ± 0.94	21.27 ^A^ ± 0.66	35.07 ^A^ ± 0.14	39.75 ^A^ ± 0.73
Late Apoptotic	3.00 ± 1.32	17.00 ± 0.23	12.00 ^a^ ± 2.35	26.13 ^bA^ ± 0.57	21.60 ^aA^ ± 2.78	15.75 ^b^ ± 1.89
Dead	0.27 ± 0.14	0.60 ± 0.00	1.80 ± 0.33	0.07 ± 0.09	1.40 ± 0.05	0.30 ± 0.05
	MDA-MB-231 cell line					
Live	73.60 ± 2.26	60.80 ± 3.34	37.00 ^aB^ ± 0.75	55.55 ^b^ ± 2.97	17.75 ^aB^ ± 0.28	30.15 ^bB^ ± 6.70
Total apoptotic	26.40 ± 2.45	39.20 ± 3.34	62.90 ^aB^ ± 0.90	44.45 ^b^ ± 3.20	82.20 ^aB^ ± 1.27	69.75 ^bB^ ± 6.93
Early Apoptotic	23.20 ± 2.31	35.60 ± 2.92	52.90 ^aB^ ± 0.71	39.30 ^bB^ ± 4.00	42.20 ^aB^ ± 1.56	58.00 ^bB^ ± 4.95
Late Apoptotic	3.20 ± 0.14	3.60 ± 0.42	10.00 ± 1.61	5.15 ^B^ ± 0.80	40.00 ^aB^ ± 0.28	11.75 ^b^ ± 1.98
Dead	0.00 ± 0.00	0.00 ± 0.00	0.10 ± 0.04	0.00 ± 0.00	0.05 ± 0.02	0.10 ± 0.23

UC—untreated control, STS—staurosporine, YS—native juice from young shoots of beetroot, R—native juice from root of beetroot, YS d+a—juice from young shoots of beetroot subjected to in vitro gastrointestinal digestion and absorption, R d+a—juice from root of beetroot subjected to in vitro gastrointestinal digestion and absorption, ^a, b^—values marked in rows with different letters differ statistically significantly (*p* ≤ 0.05) for two pairs of columns (YS vs. R and YS d+a vs. R d+a), ^A, B^—values marked in columns with different letters differ statistically significantly (*p* ≤ 0.05) between the same states of both analyzed cell lines.

**Table 5 ijms-24-06889-t005:** Comparison of BCL-2 protein inactivation in cancer cells [%] using flow cytometry (Muse^®^ Cell Analyzer).

MCF-7 Cell Line						
BCL-2 Protein	UC	STS	YS	R	YS d+a	R d+a
	Mean ± SD					
Activated	79.30 ± 0.50	23.60 ± 0.23	7.40 ^aA^ ± 0.60	68.50 ^b^ ± 0.55	8.60 ^aA^ ± 0.02	23.60 ^bA^ ± 0.08
Inactivated	20.40 ± 0.25	76.40 ± 0.45	92.60 ^aA^ ± 1.88	31.30 ^b^ ± 0.57	91.40 ^aA^ ± 0.27	76.40 ^bA^ ± 1.55
No expression	0.10 ± 0.01	0.00 ± 0.00	0.00 ± 0.00	0.20 ± 0.00	0.00 ± 0.00	0.20 ± 0.00
MDA-MB-231 Cell Line						
Activated	94.30 ± 5.86	19.70 ± 0.95	16.30 ^aB^ ± 4.05	68.60 ^b^ ± 0.45	2.40 ^aB^ ± 0.25	45.20 ^bB^ ± 6.55
Inactivated	5.40 ± 0.20	80.30 ± 8.90	83.50 ^aB^ ± 2.55	31.10 ^b^ ± 0.25	97.50 ^aB^ ± 2.55	54.50 ^bB^ ± 8.55
No expression	0.30 ± 0.02	0.00 ± 0.00	0.20 ± 0.00	0.30 ± 0.00	0.10 ± 0.03	0.30 ± 0.05

UC—untreated control, STS—staurosporine, YS—native juice from young shoots of beetroot, R—native juice from root of beetroot, YS d+a—juice from young shoots of beetroot subjected to in vitro gastrointestinal digestion and absorption, R d+a—juice from root of beetroot subjected to in vitro gastrointestinal digestion and absorption. ^a, b^—values marked in rows with different letters differ statistically significantly (*p* ≤ 0.05) for two pairs of columns (YS vs. R and YS d+a vs. R d+a). ^A, B^—values marked in columns with different letters differ statistically significantly (*p* ≤ 0.05) between the same states of both analyzed cell lines.

**Table 6 ijms-24-06889-t006:** Comparison of caspase activation capacity of the tested juices in cancer cells [%] using flow cytometry (Muse^®^ Cell Analyzer).

	MCF-7 Cell Line					
Cell	UC	STS	YS	R	YS d+a	R d+a
	Mean ± SD					
Live	88.40 ± 2.56	61.20 ± 1.48	32.70 ^aA^ ± 0.38	68.55 ^bA^ ± 2.12	27.20 ^aA^ ± 0.48	65.20 ^bA^ ± 2.40
Caspase+	9.80 ± 0.31	13.65 ± 1.29	23.55 ± 0.43	22.55 ^A^ ± 2.20	23.10 ^aA^ ± 1.27	32.70 ^bA^ ± 3.65
Caspase+/Dead	1.45 ± 0.15	24.60 ± 0.24	43.50 ^aA^ ± 1.18	8.55 ^bA^ ± 0.27	49.60 ^aA^ ± 0.56	1.95 ^bA^ ± 1.96
Total caspase	11.25 ± 1.27	38.25 ± 1.46	67.05 ^aA^ ± 1.60	31.10 ^bA^ ± 1.20	72.70 ^a^ ± 1.37	34.65 ^bA^ ± 3.38
Dead	0.35 ± 0.02	0.55 ± 0.03	0.25 ± 0.03	0.35 ± 0.01	0.10 ^A^ ± 0.00	0.15 ± 0.02
	MDA-MB-231 cell line					
Live	93.55 ± 0.95	46.30 ± 0.78	73.00 ^aB^ ± 1.56	52.85 ^bB^ ± 1.57	21.70 ^aB^ ± 2.47	40.00 ^bB^ ± 0.78
Caspase+	5.35 ± 0.07	49.70 ± 5.51	22.90 ^a^ ± 1.34	28.95 ^bB^ ± 0.46	49.20 ^aB^ ± 4.56	39.90 ^bB^ ± 0.07
Caspase+/Dead	1.00 ± 0.85	3.85 ± 7.67	3.90 ^aB^ ± 0.11	18.05 ^bB^ ± 0.11	24.40 ^aB^ ± 2.02	18.35 ^bB^ ± 0.85
Total caspase	6.35 ± 0.92	53.55 ± 1.24	26.80 ^aB^ ± 1.45	47.00 ^bB^ ± 0.57	73.60 ^a^ ± 2.19	58.25 ^bB^ ± 0.78
Dead	0.10 ± 0.04	0.15 ± 0.19	0.20 ± 0.11	0.15 ± 0.01	4.70 ^B^ ± 0.11	1.75 ± 0.11

UC—untreated control, STS—staurosporine, YS—native juice from young shoots of beetroot, R—native juice from root of beetroot, YS d+a—juice from young shoots of beetroot subjected to in vitro gastrointestinal digestion and absorption, R d+a—juice from root of beetroot subjected to in vitro gastrointestinal digestion and absorption, ^a, b^—values marked in rows with different letters differ statistically significantly (*p* ≤ 0.05) for two pairs of columns (YS vs. R and YS d+a vs. R d+a), ^A, B^—values marked in columns with different letters differ statistically significantly (*p* ≤ 0.05) between the same rows of both analyzed cell lines.

**Table 7 ijms-24-06889-t007:** Comparison of expression levels of selected genes associated with apoptosis and the cell cycle in cancer cells.

	MCF-7 Cell Line			
Type of Juice	YS	R	YS d+a	R d+a
Gene	Mean ± SD			
*AIFM1*	2.05 ± 0.04	1.46 ^A^ ± 0.08	2.17 ^A^ ± 0.11	1.36 ^A^ ± 0.41
*AKT1*	1.26 ± 0.02	1.19 ± 0.00	−1.97 ± 0.03	−2.51 ± 0.33
*APAF1*	4.12 ^A^ ± 0.10	3.31 ± 0.22	5.05 ^aA^ ± 0.20	3.61 ^b^ ± 0.36
*BAD*	2.03 ± 0.28	1.02 ± 0.09	1.84 ^A^ ± 0.05	1.76 ^A^ ± 0.48
*BBC3*	4.68 ^cA^ ± 0.37	3.64 ^c^ ± 0.16	3.30 ^a^ ± 0.28	1.00 ^b^ ± 0.01
*BCL2*	−1.24 ± 1.02	−1.67 ± 0.68	2.17 ^A^ ± 0.50	−1.68 ^A^ ± 0.01
*BID*	1.81 ± 0.02	1.12 ^A^ ± 0.10	1.33 ± 0.12	1.07 ± 0.02
*CASP3*	Nd	Nd	Nd	Nd
*CASP7*	5.07 ^a^ ± 0.32	1.96 ^b^ ± 0.16	1.51 ± 0.16	1.75 ± 0.84
*CASP8*	4.61 ^aA^ ± 0.09	2.36 ^bA^ ± 0.27	3.16 ^c^ ± 0.29	2.16 ^d^ ± 0.20
*DIABLO*	1.54 ± 0.09	1.20 ± 0.07	2.55 ^c^ ± 0.17	1.48 ^d^ ± 0.03
*FADD*	1.68 ± 0.22	1.52 ^A^ ± 0.13	1.10 ^A^ ± 0.02	1.28 ^A^ ± 0.15
*FAS*	3.36 ^a^ ± 0.28	1.01 ^b^ ± 0.22	−1.42 ^a^ ± 0.56	1.10 ^b^ ± 0.01
*MYC*	1.58 ± 0.09	1.48 ^A^ ± 0.03	1.89 ± 0.03	1.78 ± 0.06
*NFKB1*	3.17 ^a^ ± 0.07	1.40 ^bA^ ± 0.05	1.40 ^aA^ ± 0.11	1.10 ^bA^ ± 0.08
*PHLPP1*	1.40 ± 0.28	1.68 ± 0.56	−1.35 ^a^ ± 0.70	1.19 ^b^ ± 1.07
*TP53*	1.70 ± 0.10	1.10 ^A^ ± 0.07	1.55 ± 0.07	1.31 ± 0.07
	MDA-MB-231 cell line			
*AIFM1*	2.14 ^a^ ± 0.05	7.75 ^bB^ ± 0.02	3.72 ^B^ ± 0.11	3.85 ^B^ ± 0.02
*AKT1*	−1.07 ± 0.04	−1.27 ± 0.35	−2.23 ± 0.12	−2.16 ± 0.52
*APAF1*	2.45 ^B^ ± 0.05	2.89 ± 0.02	2.10 ^B^ ± 0.15	2.66 ± 0.55
*BAD*	1.24 ± 0.10	1.34 ± 0.09	4.21 ^B^ ± 0.07	3.29 ^B^ ± 0.41
*BBC3*	1.23 ^cB^ ± 0.38	2.19 ^d^ ± 1.20	2.65 ^a^ ± 0.48	1.07 ^b^ ± 0.59
*BCL2*	Nd	Nd	1.01 ^B^ ± 0.26	1.33 ^B^ ± 0.24
*BID*	1.32 ^a^ ± 0.74	2.91 ^bB^ ± 0.22	1.57 ± 0.04	1.14 ± 0.01
*CASP3*	Nd	Nd	Nd	Nd
*CASP7*	Nd	Nd	Nd	Nd
*CASP8*	−5.01 ^aB^ ± 1.18	−1.56 ^bB^ ± 0.07	3.38 ^c^ ± 0.06	2.44 ^d^ ± 0.01
*DIABLO*	1.99 ± 0.06	2.25 ± 0.03	1.43 ^c^ ± 0.07	2.50 ^d^ ± 0.25
*FADD*	−1.92 ^c^ ± 0.02	−2.65 ^dB^ ± 0.53	5.30 ^aB^ ± 0.04	−3.00 ^bB^ ± 0.70
*FAS*	Nd	Nd	Nd	Nd
*MYC*	1.00 ^a^ ± 0.41	3.54 ^bB^ ± 0.08	1.45 ± 0.05	1.38 ± 0.02
*NFKB1*	2.02 ^a^ ± 0.21	6.40 ^bB^ ± 0.30	5.48 ^aB^ ± 0.07	3.96 ^bB^ ± 0.35
*PHLPP1*	−1.00 ^a^ ± 0.00	1.35 ^b^ ± 0.03	−1.20 ± 0.55	1.25 ± 0.26
*TP53*	2.81 ± 0.09	2.60 ^B^ ± 0.03	1.45 ± 0.09	1.35 ± 0.15

YS—native juice from young shoots of beetroot, R—native juice from root of beetroot, YS d+a—juice from young shoots of beetroot subjected to in vitro gastrointestinal digestion and absorption, R d+a—juice from root of beetroot subjected to in vitro gastrointestinal digestion and absorption, Nd—expression was not detected. ^a, b^—values marked in rows with different letters differ statistically significantly (*p* ≤ 0.05) between two pairs of columns (YS vs. R and YS d+a vs. R d+a), ^c, d^—values marked with different letters in the rows indicate a trend (*p* ≤ 0.1) between the two pairs of columns (YS vs. R and YS d+a vs. R d+a), ^A, B^—values marked in columns with different letters between both analyzed cell lines differ statistically significantly *p* ≤ 0.05. Nd—not detected.

## Data Availability

Not applicable.

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
