# Peer review of "Young Shoots of Red Beet and the Root at Full Maturity Inhibit Proliferation and Induce Apoptosis in Breast Cancer Cell Lines"

_ijms, 2023, doi:10.3390/ijms24086889_

Round 1

Reviewer 1 Report

The paper requires minor spell check (e.g. line 641 the word digest-ed). It would be better to improve the quality of figures 3 and 4: they are quite blurry and not easy to read.

Author Response

Ewelina Piasna-SÅ‚upecka,                                                23th March, 2022, Krakow

University of Agriculture in Krakow

Faculty of Food Technology

Department of Human Nutrition and Dietetics

Balicka 122, 30-149 Krakow

tel. +48126624817

Dear Reviewer,

we would like to thank for time and efforts put into thorough reading and the review of the manuscript as well as for giving comments and suggestions on how to improve it. We have found all comments valuable and have implemented the corrections according to suggestions. We hope all the changes and improvements we introduced to the revised manuscript will meet journal requirements and Editor acceptance.

Modified parts in the revised manuscript are marked using the "Track Changes" function in Microsoft Word. Below, please find our detailed answers to the Reviewers’ comments and suggestions.

Detailed responses to Reviewer:

In the review: The paper requires minor spell check (e.g. line 641 the word digest-ed).

Our answer: Thank you for this comment. The spell has been corrected.

In the review: It would be better to improve the quality of figures 3 and 4: they are quite blurry and not easy to read.

Our answer: Thank you for this comment. The quality of figures 3 and 4 have been improved.

We hope that all the changes and improvements are satisfied. We also declare our willingness to make further improvements if such are identified by the Editor or Reviewers.

We declare that the manuscript has not been submitted for publication elsewhere.
All co-authors have contributed to this article and all agree to submit it into the International Journal of Molecular Sciences. There are no conflicts of interests.

We would be grateful for the acceptance of our manuscript for publication.

Yours sincerely,

On behalf of the Authors,

Ewelina Piasna-SÅ‚upecka

Reviewer 2 Report

The manuscript presents valuable study on the anti-proliferative and apoptotic effects of red beet juices (young shoots and root at full maturity) on hormone-dependent and hormone-independent breast cancer cell lines. Chemical composition of both juices was also determined.

The manuscript is well-structured, written in a scientifically sound manner. The material and methods are described in detail. The results are presented in a systematic manner and the discussion of the obtained results is extensive.

Remarks are listed below:

The paper would benefit from proofreading by a native English speaker in order to improve grammatical mistakes and paraphrase certain sentences, such as in line 67 – use either „the content of polyphenols“ or „quantitative polyphenol profile“ instead of „the content of the quantitative polyphenol profile“; line 77-79; line 524-526; line 611-616.

In Table 1, the content of vanillic acid, syringic acid and sinapinic acid is presented, among others. However, in lines 75-76, 79, vanillin, „syring“ and sinapine are stated. The names of compounds should be written correctly.

Line 453 – it should be p-coumaric acid

Scientific name of species should be written italic (line 542, 546)

Author Response

Ewelina Piasna-SÅ‚upecka,                                                23th March, 2022, Krakow

University of Agriculture in Krakow

Faculty of Food Technology

Department of Human Nutrition and Dietetics

Balicka 122, 30-149 Krakow

tel. +48126624817

Dear Reviewer,

we would like to thank for time and efforts put into thorough reading and the review of the manuscript as well as for giving comments and suggestions on how to improve it. We have found all comments valuable and have implemented the corrections according to suggestions. We hope all the changes and improvements we introduced to the revised manuscript will meet journal requirements and Editor acceptance.

Modified parts in the revised manuscript are marked using the "Track Changes" function in Microsoft Word. Below, please find our detailed answers to the Reviewers’ comments and suggestions.

Detailed responses to Reviewer:

In the review: The paper would benefit from proofreading by a native English speaker in order to improve grammatical mistakes and paraphrase certain sentences, such as in line 67 – use either „the content of polyphenols“ or „quantitative polyphenol profile“ instead of „the content of the quantitative polyphenol profile“; line 77-79; line 524-526; line 611-616.

Our answer: Thank you for this comment. The grammatical mistakes and paraphrase certain sentences have been improved by native speaker.

In the review: In Table 1, the content of vanillic acid, syringic acid and sinapinic acid is presented, among others. However, in lines 75-76, 79, vanillin, „syring“ and sinapine are stated. The names of compounds should be written correctly.

Our answer: Thank you for this comment. The names of compounds have been corrected.

In the review: Line 453 – it should be p-coumaric acid.

Our answer: Thank you for this comment. However, the results presented in Table 1 show that “In young shoots of beetroot, the major compound was sinapinic acid from the hy-droxycinnamic group, whereas in the root – it was catechin belonging to the class of flavanols.” Therefore, no change has been made.

In the review: Scientific name of species should be written italic (line 542, 546).

Our answer: Thank you for this comment. We fully agree. There is no species name in the line 542, 546. It was probably about the name of Brassica oleracea var. capitata f. alba and Brassica oleracea var. capitata f. rubra. The name of species has been corrected.

We hope that all the changes and improvements are satisfied. We also declare our willingness to make further improvements if such are identified by the Editor or Reviewers.

We declare that the manuscript has not been submitted for publication elsewhere.
All co-authors have contributed to this article and all agree to submit it into the International Journal of Molecular Sciences. There are no conflicts of interests.

We would be grateful for the acceptance of our manuscript for publication.

Yours sincerely,

On behalf of the Authors,

Ewelina Piasna-SÅ‚upecka

Reviewer 3 Report

Although this manuscript clearly shows the antiproliferative and apoptotic effects of polyphenol profile in the juice of young shoots of the beetroot on breast cancer cell lines MCF-7 and MDA-MB-231, I have some comments as follows:  

Major comments:

Images of western blots have likely been faked.  The authors should require the whole uncropped western blot images for all replicates used in the quantitative analysis.

Minor comments:

1. In the Abstract section, I suggest clearly pinpointing the regulatory pathways of the antiproliferative and apoptotic effects.

2. Although the author tried to provide the signal peak of polyphenols, Figures 1, and 2 are still not clear enough. They should that different polyphenols standards have their own single peaks and the respective RT of polyphenols must be clearly displayed in Table 1.

3. There were too many references cited, and a selective citation was required.

Author Response

Ewelina Piasna-SÅ‚upecka,                                               23th March, 2023, Krakow

University of Agriculture in Krakow

Faculty of Food Technology

Department of Human Nutrition and Dietetics

Balicka 122, 30-149 Krakow

tel. +48126624817

Dear Reviewer,

we would like to thank for time and efforts put into thorough reading and the review of the manuscript as well as for giving comments and suggestions on how to improve it. We have found all comments valuable and have implemented the corrections according to suggestions. We hope all the changes and improvements we introduced to the revised manuscript will meet journal requirements and Editor acceptance.

Modified parts in the revised manuscript are marked using the "Track Changes" function in Microsoft Word. Below, please find our detailed answers to the Reviewers’ comments and suggestions.

Detailed responses to Reviewer:

In the review: Images of western blots have likely been faked. The authors should require the whole uncropped western blot images for all replicates used in the quantitative analysis.

Our answer: Thank you for this comment. In successive Western blot experiments, the samples were arranged in two different ways by 2 persons. The reason for cutting out the Western blot images was related to the need to arrange them according to the logically presented order of the discussed results.

In the review: In the Abstract section, I suggest clearly pinpointing the regulatory pathways of the antiproliferative and apoptotic effects.

Our answer: Thank you for this comment. In the Abstract section the regulatory pathways of the antiproliferative and apoptotic effects have been clearly pinpointed. The 200-word limit in this chapter only allows to add a few crucial words.

In the review: Although the author tried to provide the signal peak of polyphenols, Figures 1, and 2 are still not clear enough. They should that different polyphenols standards have their own single peaks and the respective RT of polyphenols must be clearly displayed in Table 1.

Our answer: Due to the large number of the determined compounds and the detection at different wavelengths (additionally because of the huge number and volume of presented results), the authors have resigned from presenting all determined compounds at all wavelengths on the chromatograms. The authors afraid that this could disturb the presented results. The respective RT of polyphenols have been clearly displayed in Table 1. If all files are needed, then they will be added. The work presents only selected sample files, as in many other published.

In the review: There were too many references cited, and a selective citation was required.

Our answer: Thank you for this comment. Excess citations have been removed and selective citations have been left.

We hope that all the changes and improvements are satisfied. We also declare our willingness to make further improvements if such are identified by the Editor or Reviewers.

We declare that the manuscript has not been submitted for publication elsewhere.
All co-authors have contributed to this article and all agree to submit it into the International Journal of Molecular Sciences. There are no conflicts of interests.

We would be grateful for the acceptance of our manuscript for publication.

Yours sincerely,

On behalf of the Authors,

Ewelina Piasna-SÅ‚upecka

Round 2

Reviewer 3 Report

Even though the original picture is attached, Lane R is still suspected of being fake. There are obvious collage marks.

Author Response

Ewelina Piasna-SÅ‚upecka,                                                29th March, 2023, Krakow

University of Agriculture in Krakow

Faculty of Food Technology

Department of Human Nutrition and Dietetics

Balicka 122, 30-149 Krakow

tel. +48126624817

Dear Reviewer,

we are grateful for comment and review of the article one more time. The manuscript has been improved according to your suggestions. We hope that all the changes and improvements we have introduced into the revised manuscript will meet your and Editor acceptance.

Modified parts in the revised manuscript and supplement are marked using the "Track Changes" function in Microsoft Word. Below, please find our detailed answers to the Reviewers’ comments and suggestions.

Detailed responses to Reviewer:

In the review: Even though the original picture is attached, Lane R is still suspected of being fake. There are obvious collage marks.

Our answer: Thank you for this comment. We sincerely apologize if we have made you confused and doubtful. We would like to explain why we didn't include the original western blot images in the right way. This is the first time we've been asked for this. We have never before sent original photos. We hope that these original photos will be enough. Thank you for the opportunity to explain that all. I assure you that no changes have been made that would affect the results. I kindly ask you to consider accepting this manuscript based on the right results I have already presented.

We hope that all the changes and improvements are satisfied. We also declare our willingness to make further improvements if such are identified by the Editor or Reviewers.

We declare that the manuscript has not been submitted for publication elsewhere.
All co-authors have contributed to this article and all agree to submit it into the International Journal of Molecular Sciences. There are no conflicts of interests.

We would be grateful for the acceptance of our manuscript for publication.

Yours sincerely,

On behalf of the Authors,

Ewelina Piasna-SÅ‚upecka

Round 3

Reviewer 3 Report

The original images must be attached to supplemental data and marked for responsive lanes in every group.

Author Response

Ewelina Piasna-SÅ‚upecka,                                                29th March, 2023, Krakow

University of Agriculture in Krakow

Faculty of Food Technology

Department of Human Nutrition and Dietetics

Balicka 122, 30-149 Krakow

tel. +48126624817

Dear Reviewer,

we would like to thank for time and efforts put into thorough reading and the review of the manuscript one more time as well as for giving comment and suggestion on how to improve it. We have found all comments valuable and have implemented the corrections according to suggestions. We hope all the changes and improvements we introduced to the revised manuscript will meet journal requirements and Editor acceptance.

Modified parts in the revised manuscript are marked using the "Track Changes" function in Microsoft Word. Below, please find our detailed answers to the Reviewers’ comments and suggestions.

Detailed responses to Reviewer:

In the review: The original images must be attached to supplemental data and marked for responsive lanes in every group.

Our answer: Thank you for these suggestions. The original images were attached to a supplemental file and signed like Supplementary Figure S10 and Supplementary Figure S12. All lanes in every group have been signed. The new supplementary figures were also included in the manuscript. As we wrote before, the reason for cutting out the Western blot images in the manuscript was related to the need to arrange them according to the logically presented order of the discussed results, which might have made you doubt. We would like to one more time apologize for that. Due to the fact that now we have the ability to change the "cut version", in which the order has been changed according to the discussed results, we kindly ask for suggestions on which version will be appropriate. The original version has a different order signed.

We hope that all the changes and improvements are satisfied. We also declare our willingness to make further improvements if such are identified by the Editor or Reviewers.

We declare that the manuscript has not been submitted for publication elsewhere.
All co-authors have contributed to this article and all agree to submit it into the International Journal of Molecular Sciences. There are no conflicts of interests.

We would be grateful for the acceptance of our manuscript for publication.

Yours sincerely,

On behalf of the Authors,

Ewelina Piasna-SÅ‚upecka

Round 4

Reviewer 3 Report

I have no additional comments.